behaviour, cognition

causal inference, unseen causes, preschoolers, capuchin monkeys

**Author for correspondence:**
Zeynep Civelek
e-mail: zc8@st-andrews.ac.uk

# What happened? Do preschool children and capuchin monkeys spontaneously use visual traces to locate a reward?

Zeynep Civelek[1], Christoph J. Völter[1,2] and Amanda M. Seed[1]

[1]School of Psychology and Neuroscience, University of St Andrews, St Andrews, UK
[2]Messerli Research Institute, University of Veterinary Medicine Vienna, Medical University of Vienna, University of Vienna, Vienna, Austria

ZC, 0000-0001-8483-3752; CJV, 0000-0002-8368-7201; AMS, 0000-0002-3867-3003

The ability to infer unseen causes from evidence is argued to emerge early in development and to be uniquely human. We explored whether preschoolers and capuchin monkeys could locate a reward based on the physical traces left following a hidden event. Preschoolers and capuchin monkeys were presented with two cups covered with foil. Behind a barrier, an experimenter (E) punctured the foil coverings one at a time, revealing the cups with one cover broken after the first event and both covers broken after the second. One event involved hiding a reward, the other event was performed with a stick (order counterbalanced). Preschoolers and, with additional experience, monkeys could connect the traces to the objects used in the puncturing events to find the reward. Reversing the order of events perturbed the performance of 3-year olds and capuchins, while 4-year-old children performed above chance when the order of events was reversed from the first trial. Capuchins performed significantly better on the ripped foil task than they did on an arbitrary test in which the covers were not ripped but rather replaced with a differently patterned cover. We conclude that by 4 years of age children spontaneously reason backwards from evidence to deduce its cause.

## 1. Introduction

A scientist presented with a skin rash, a child discovering her toy box open and an animal coming across footprints all face a similar challenge. There is a gap in their knowledge about what caused these visual traces and detecting the culprit is desirable. While there is no doubt that adults look for and detect causes given incomplete information, there is less consensus on when the ability to think about causality in a theory-like way emerges in human development [1–3]. Whether or not this skill exists at all in non-human animals is also a matter of debate [4,5]. These issues mainly originate from the fact that making inferences about an unseen cause is not the only way of responding appropriately to indirect evidence.

If one repeatedly encounters certain events in close spatio-temporal proximity (e.g. a jaguar walking past and leaving footprints behind), one may learn to use the latter as a cue to the former without causal reasoning [6]. Such statistical learning is common to humans, even in infancy [7] as well as several species of non-human animals [8]. In addition, humans, once they have language, can use testimony from others [9,10]. Given these alternative routes to responding to unseen causes, it has been a challenge to distinguish them in developmental and comparative research.

In this paper, we aim to explore the developmental and phylogenetic origins of the ability to use evidence to detect a hidden cause. From an evolutionary perspective, some authors suggested that humans are the only species capable of representing and reasoning about causality [4], while others contend that we may share some mechanisms for dealing with natural causal structures, in order

to infer the location of food or when using tools [5]. Nevertheless, there is general agreement that some components of human causal reasoning represent a significant departure from anything present in other animals. The tendency to seek explanations for unseen events is one candidate, because it might require a kind of thinking about past possibilities scaffolded through cultural and linguistic input over human development [11,12]. From a developmental perspective, the timing of emergence of this ability is therefore important, as it helps us to determine whether humans have a natural tendency to look for and infer causal relations in their environments early in life, perhaps even from birth [13,14]. If so, its development may follow a single trajectory into adulthood and the difference between younger and older children and adults may be explained by other domain-general components (e.g. information processing capacity) and expanding causal knowledge. Alternatively, early reasoning abilities may undergo a significant developmental change in childhood due to the cultural input from others with the development of language [1,10].

In the last few decades, developmental research has found extensive evidence for various kinds of causal inference in preschool children. They used information about physical mechanism not just spatio-temporal contiguity [13,15,16], inferred whether objects had causal power based on conditional probabilities and interventions [17,18], expected to find a hidden cause when they were presented with an unexpected outcome [19,20] and came up with informative tests that could uncover ambiguity [21]. These studies often demonstrated a developmental difference between 3 and 4 years of age [13,15,18,22].

However, the existence of a causal relationship and even the possibility that it might be unseen was provided in the verbal framing of these tasks (e.g. 'Which blower made the candle go?', 'Why are the puppets moving together? Is it X, Y or something else?'). These prompts did not specify the exact cause, but they testified that the covariations children observed were causal [10]. Such prompts have been shown to boost preschoolers' performance in a number of studies [23–27]. To find out more about how children diagnose causes from evidence alone, it is desirable to provide less explicit verbal instructions. This would also facilitate comparisons between children and non-human primates, to examine the evolution of this ability.

From a comparative perspective, some researchers have argued that non-human primates cannot extract causal relations from spatio-temporal regularities [4,28]. Limitations in performance in some tasks provide support for this interpretation [25,29–33]. By contrast, others assert that non-human primates can extract causal knowledge beyond perceptual information if the testing situation does not overload other cognitive resources [5]. This has been shown in research that compared animals' learning in two contexts: (i) where the evidence was caused by the food, and (ii) where the evidence correlated with the food's presence but the relation was arbitrary [34–37]. For example, when food was hidden on a balance beam, apes preferred the end of the beam that moved down with food's weight but they did not have a preference for the lower beam when an experimenter pushed it down [35].

In this study, we asked: do preschoolers and non-human primates spontaneously use visual traces to locate a reward? If so, is this ability driven by inferring causal relations or

associative learning of task-specific features? We tested capuchin monkeys; extractive foragers with a high proclivity to manipulate objects. Capuchins mostly feed on seasonal fruits; during the dry season they locate unseen food inside tree barks, husks and shells, and underground nests [38]. In such unstable environments, the ability to make inferences about hidden causes based on incomplete observational evidence (e.g. inferring that a nut is empty based on wormholes on its shell) would allow the monkey to go beyond learning only about causal relations that they can manipulate (e.g. tapping on the nut to infer its fullness or weighing a tool before using it to crack a nut) [39]. There is evidence that capuchin monkeys use the sound of a baited shaken cup to locate a reward, though they failed to reason by exclusion when an empty cup was shaken instead [36,40]. They have been reported to explore the weight of a nut or a stone tool before opening or using it; however, the role of extensive prior experience was not eliminated as an associative explanation in these cases [41,42]. To our knowledge, this is the first study that assessed reasoning about unseen causes based on indirect visual evidence in capuchin monkeys.

The current task required participants to locate a reward in one of two cups covered with foil, based on the appearance of rips in the foil covers following two sequential, partially hidden events. Previous research shows that children at least by 3 years of age can update their knowledge (i.e. about the location of the reward) as they receive direct visual information about events in a certain order [43,44]. In our task, they will be required to do so using indirect visual information. As for the capuchin monkeys, there is also evidence for the capacity to represent hidden objects: they expected to find the same number/type of rewards in an apparatus as they had previously seen being hidden [45]. In the current study, E covered two empty cups at the beginning of each trial with foil and showed participants that the covers were intact. Behind a barrier, she punctured the foil coverings one at a time, revealing the cups with one cover broken after the first event and both covers broken after the second (figure 1). In one event she first showed that she had a reward in her hand, then punctured one cup with the reward behind the barrier and dropped it in that cup, revealing an empty hand. In the other event, she punctured the foil covering the other cup with a pen/stick. The order of these events was counterbalanced across participants. Participants could locate the reward in the correct cup if they inferred the cause of the ripped foil but also if they learnt an arbitrary rule such as choosing the first/last change they saw. The latter route to success was tested in the Transfer phase, in which the order of events was reversed. We hypothesized that if the participants made inferences, they should search for the reward in the correct cup without a drop in performance in the Transfer phase. However, if the participants used an arbitrary rule to solve the task in the Test phase, we expected them to perform at or below chance level in the Transfer phase.

# 2. Experiment 1—children

## (a) Methods

### (i) Participants

Eighty-three 3–5 year olds ($M_{age}$ = 52.10, s.d.$_{age}$ = 9.58) participated. Six additional children were tested but did not complete the task. Age and sex were counterbalanced as

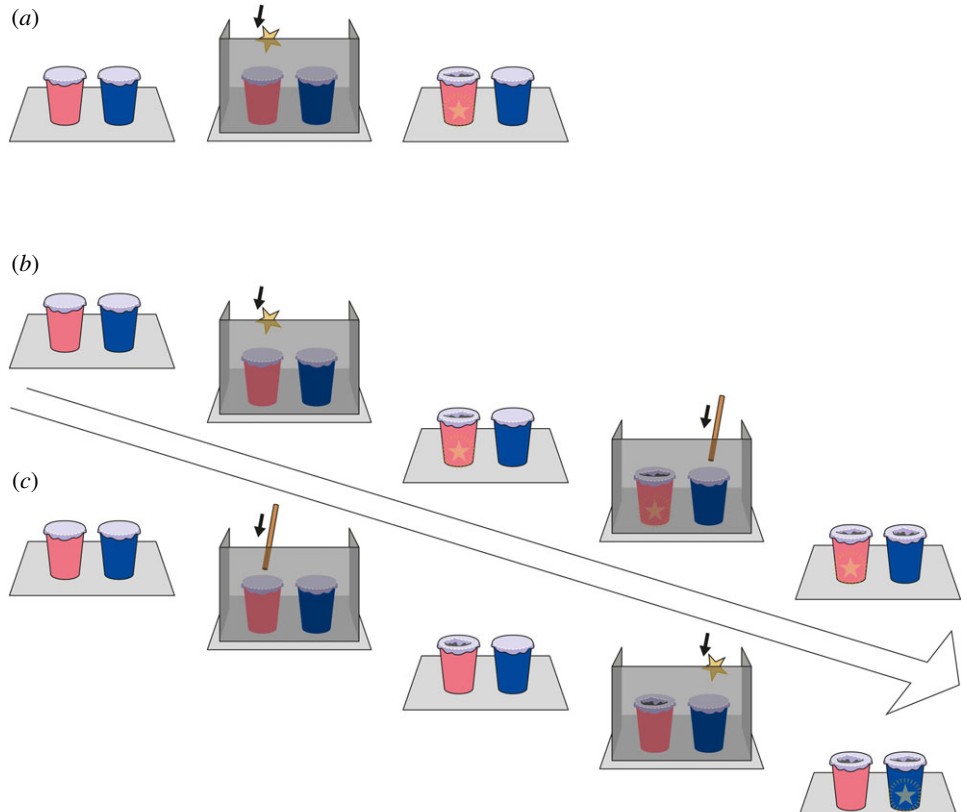

**Figure 1.** A visual illustration of the (*a*) Pre-test, (*b*) Test and (*c*) Transfer phases of Experiment 1. The illustration shows a sticker–pen trial for the Test phase and a pen–sticker trial for the Transfer phase (note that the order of conditions was counterbalanced across participants). The star represents the reward, and the stick represents the pen. The barrier and the cups are depicted as semi-transparent for the reader but these were opaque in the study. The diagonal arrow shows the temporal progression of the trial. (Online version in colour.)

much as possible (electronic supplementary material, table S1). Our target sample size was 32 children in each age group (stated in the ethics application). Due to the opportunistic sampling approach in nurseries, time constraints, high drop-out rates in the pre-test phase in 3-year olds ($N = 11$) and the lower number of 5-year olds in nurseries we only reached our target sample size with 4-year olds. No analysis was conducted prior to the end of data collection. The study was ethically approved by University of St Andrews Teaching and Research Ethics Committee and informed consent was taken from parents/guardians.

## (b) Materials

Two different coloured plastic cups (pink and blue), were placed on a table apart (approx. 20 cm) from each other. The cups were covered with pieces of aluminium foil. Stickers were used as rewards and a brightly coloured pen was used as the non-reward object to poke holes in the foil covers. A barrier was used to conceal the ripping events.

### (i) Procedure

*Pre-test.* E covered the cups with foil and placed the barrier in front of them. She picked up a sticker from the table while the child watched and put it in one of the cups. The action of forcing the sticker through the foil was completely hidden from the child's perspective, though they could hear the foil being punctured (figure 1). When the barrier was removed, E moved the cups apart, tilted them slightly and asked, 'Where do you think the sticker is?' If the child pointed to the correct cup, E removed the foil and gave the sticker to the child. She then removed the foil covering the other cup

to reveal that it was empty. If the child did not find the sticker, E showed the contents of both cups and prepared for the next trial. The criterion was the first two consecutive trials correct or four trials correct in a maximum of five trials. Otherwise, the participant was considered a drop-out (15 children: eleven 3-year olds and four 4-year olds). Of the 68 participants who passed the pre-test, 64 of them passed it within two trials and four needed five trials ($M = 2.18$, s.d. $= 0.71$).

*Test.* After the pre-test, E said 'This game was too easy for you! Shall we make it more fun?' and introduced the new task. 'Now the game is slightly different. I will cover the cups again and I will put these here (as she put a pen and a sticker on the table where the child could see). Now watch me till the end and then point to the cup you think the sticker is in! Ready?'

The participants saw E rip the two cups in one of the two orders:

(1) *Sticker–pen*: E first picked up the sticker and used it to puncture the foil covering one of the cups behind the barrier and she dropped the sticker in the cup. The pen remained in view. Then she removed the barrier to reveal the foil covering one of the cups was ripped. Next, she put the barrier back saying, 'Keep watching' and took the pen to rip the other cup. Then, she placed the pen back on the table, removed the barrier revealing both cups were ripped and asked 'Now, where is the sticker?'

(2) *Pen–sticker*: The set-up was the same but this time E first picked up the pen and then the sticker.

After the participants received eight trials of Test, they moved on to the Transfer phase.

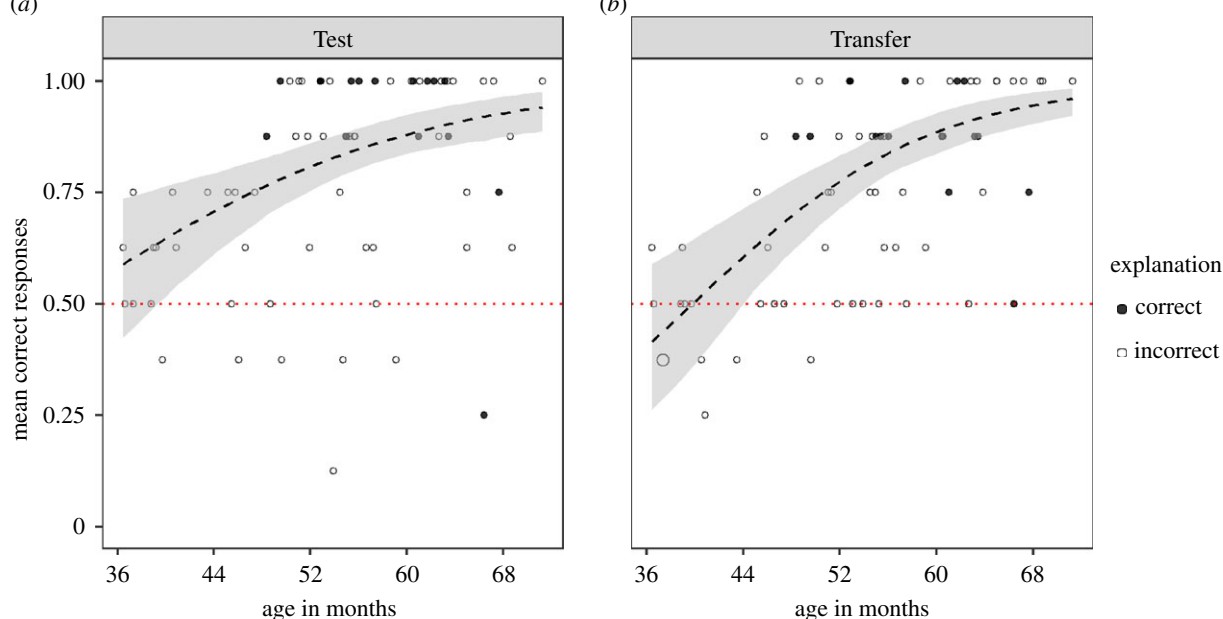

**Figure 2.** Performance of children in the Test (*a*) and Transfer (*b*) phases. Dotted line shows the chance level. Dashed line shows the model predictions, and the shaded area is the bootstrapped 95% confidence interval (with the predictor variables age, phase, trial type, trial number and sex centred). Children's responses to the explanation question at the end of the task is presented as open (incorrect) and solid (correct) circles. (Online version in colour.)

*Transfer.* In this phase, the participants received eight trials in the reverse order (e.g. if they had *sticker–pen* trials in Test, they received *pen–sticker* trials in Transfer).

Every trial in the Test and Transfer phases started with two cups covered with foil, had an intermediate stage in which the barrier was removed to show the cover on one cup was ripped and ended with both covers being ripped. The two poking events were always done on different cups and the participants could hear two puncturing sounds. The side of the blue/pink cup was randomized across participants, but stayed the same for each participant (e.g. blue always on the left). The cup on which the sticker was hidden varied in a pseudo-random fashion in each phase with the constraint that it appeared equally in both cups, and never in the same cup more than twice in succession. The pen and the sticker lay at the same side of the table throughout the game for each subject.

*Explanation question.* At the end of the task, E asked; 'How did you decide which cup to choose?' If children did not reply, E elaborated 'Sometimes the sticker was in the blue cup and sometimes in the pink cup. How did you know where it was?'

### (ii) Scoring and analysis

The first choice of the participants was scored as their response.

We analysed the data using generalized linear mixed models (GLMM) [46]; with binomial error structure and logit link function using the function glmer of the R-package lme4 [47]. For the first trial analysis in the Transfer phase, we conducted a binomial test. More details concerning the analysis can be found in the electronic supplementary material.

Children's responses to the explanation question were categorized as 'correct' and 'incorrect'. The incorrect explanations consisted of those that referred to the wrong/ improbable ways of solving the task (e.g. the sticker was put in cups in alternating fashion, 'I heard them', 'I saw through the barrier') and those that did not refer to any

strategy (e.g. 'I don't know', 'I just guessed', points to the cups). The correct explanations consisted of those that referred to the ripping of the cups (e.g. 'You smashed it when you put it in', 'It got popped').

The data for all the experiments reported in this paper can be found in the electronic supplementary material.

### (c) Results

Children performed significantly above chance in both the Test (intercept-only GLMM Estimate ± s.e.: $1.63 \pm 0.22$, $z = 7.43$, $p < 0.001$) and Transfer ($1.36 \pm 0.20$, $z = 7.07$, $p < 0.001$) phases. The full model comprising two interaction terms age*phase and trial-type*phase, with sex and trial number as fixed effects fitted the data better than the null model (likelihood-ratio test: $\chi^2_6 = 34.16$, $p < 0.001$; electronic supplementary material, table S2). The trial-type*phase interaction was not significant; however, figure 2 shows a trend for a steeper age effect in Transfer phase (age*phase: $\chi^2_1 = 2.93$, $p = 0.087$). Although the majority of children performed significantly above chance level in the Test phase, young children's (below 44 months) performance did not differ from chance in the Transfer phase.

To examine the main effects of age and phase, we fitted another model without the non-significant interactions. The reduced model also fitted the data better than the null model ($\chi^2_4 = 31.21$, $p < 0.001$, electronic supplementary material, table S3). Older children performed significantly better than younger ones (Age: $\chi^2_1 = 26.08$, $p < 0.001$; electronic supplementary material, figure S1). The children's performance was not significantly affected by the change in the order of events (phase: $\chi^2_1 = 1.61$, $p = 0.205$), trial type ($\chi^2_1 = 2.86$, $p = 0.091$) or trial number ($\chi^2_1 = 2.32$, $p = 0.128$). There was no significant difference between the performances of boys and girls ($\chi^2_1 = 0.20$, $p = 0.653$).

We analysed the first trial performance in the Transfer phase to see how children performed when the order of events was reversed for the first time. Overall, children performed significantly above chance level (47 out of 68 children chose the correct cup, binomial test: $p < 0.01$): the

majority could adapt to the change in the order of events from the first trial. Given the significant age effect on performance, we also explored children's first trial performance in the Transfer phase across ages. The GLM analysis with the predictor variable age centred was compared to the null model comprising only the intercept (electronic supplementary material, table S4). Younger children made more errors than older children ($\chi_1^2 = 9.48$, $p < 0.01$). Below 50 months of age children's performance did not differ from chance level (electronic supplementary material, figure S2).

### (i) Explanation question

Only 16 children gave the correct explanation to E's question 'How did you decide which cup to choose?' at the end of the task. Older children gave the correct explanation significantly more often than younger children (GLM: $\chi_1^2 = 5.15$, $p < 0.05$) (electronic supplementary material, table S5). None of the children below 48 months gave the correct explanation (figure 2).

## (d) Discussion

Most children could use the ripped foil spontaneously as a cue to detect the location of the hidden reward in the Pretest phase. Older preschoolers went on to distinguish between the ripped foil caused by the reward and that caused by the 'irrelevant' cause (i.e. the pen) in the Test phase. Results from the Transfer phase, overall and in the first trial, also showed that older preschoolers were not using an arbitrary rule to solve the task. We suggest that the older preschoolers inferred the likely cause of the visible traces and used this information to locate the reward. On the other hand, children's verbal explanations about how they found the sticker lagged behind their performance: the majority of the children could not provide the correct explanation. Similar differences between children's performance and causal explanations were found in a study that explored 5–10-year-old children's understanding of continuous causal processes [48].

Possible explanations for the comparatively poor performance of 3-year olds will be discussed in the general discussion.

# 3. Experiment 2—capuchin monkeys

## (a) Methods

### (i) Participants

Nineteen capuchin monkeys housed in Living Links Research Centre Edinburgh Zoo ($N_{females} = 6$, $M_{age} = 9.11$, s.d.$_{age} = 3.86$) participated. All monkeys that were available and willing to participate were tested. Nine participants got *food–stick* trials in the Test phase and *stick–food* in the Transfer phase, and 10 participants got the reverse order (electronic supplementary material, table S6). The research was conducted in accordance with the regulations of the University of St Andrews' Animal Welfare and Ethics Committee and Edinburgh Zoo.

## (b) Materials

The same as in Experiment 1. We used raisins or dates as rewards. We used a table with wheels to present the materials levelled with the bottom of the testing cubicle.

### (i) Procedure

The procedure for the monkeys differed from children in the following ways: (i) the monkeys received an additional Warm-up for familiarization. The monkeys had extensive experience in finding a reward hidden in cups [49]—so this phase was just to ensure that the novel materials (tall colourful cups) did not cause any neophobia. The monkeys were able to pass this phase with no difficulty. (ii) Monkeys received eight trials per session and the criterion to pass a phase was 14/16 trials correct (binomial test: $p = 0.001$) in two consecutive sessions. (iii) We decided (also due to our limited sample size) that if monkeys did not perform at above chance levels in the first 16 trials, they would receive further trials (electronic supplementary material, table S7) to see if they could learn the task before moving on to the last phase where we evaluated their ability to flexibly transfer to a reversed order of events.

*Warm-up.* E put a raisin in one of the cups in full view and pushed the table towards the subject to choose. The number of sessions needed to pass this phase ranged from two to 18 ($M = 7.0$, s.d. = 5.36).

*Pre-test.* Same as in Experiment 1. Monkeys could receive up to 10 sessions in total if they did not reach the criterion sooner. All the monkeys passed this phase within two to six sessions ($M = 2.94$, s.d. = 1.13). For the purposes of comparison with the child data (criterion: 4/5 correct), it is worth noting that eight out of 19 monkeys reached the criterion in the first five trials (electronic supplementary material, table S7 for a comparison).

*Test.* The same as in Experiment 1. If the participants did not reach the criterion sooner, they moved on to the Transfer phase after 10 sessions.

*Transfer.* The participants received two sessions (16 trials) in the reverse order to the Test phase.

### (ii) Scoring and analysis

Scoring was the same as Experiment 1.

In the main analyses, we used the last 16 trials of the Test phase for each monkey (as opposed to all trials) and the 16 trials (two sessions) of the Transfer phase. Since we were interested in the potential performance change from Test to Transfer, we wanted to include the monkeys' best performance in the Test phase.

The full model comprised phase (Test/Transfer), trial type (food–stick/stick–food) and their interaction; age, sex and trial number as fixed effects. Subject ID was included as the random effect as well as all possible random slope components except for the correlations between random intercepts and slopes. Comparisons to the hypothetical chance level were done in the same way as in Experiment 1. There were no issues with regards to the model stability or multicollinearity.

## (c) Results

The monkeys' performance in the first eight trials of the Test phase did not differ from chance ($0.16 \pm 0.17$, $z = 0.93$, $p = 0.35$). Only 1/19 monkeys performed above chance level in the first eight trials of the Test phase according to a two-tailed binomial test ($p = 0.001$). We therefore proceeded to give the monkeys further trials to explore their ability to learn to solve this task, and then transfer their knowledge. The number of sessions needed to pass the Test phase ranged from a minimum of two (16 trials) to a maximum

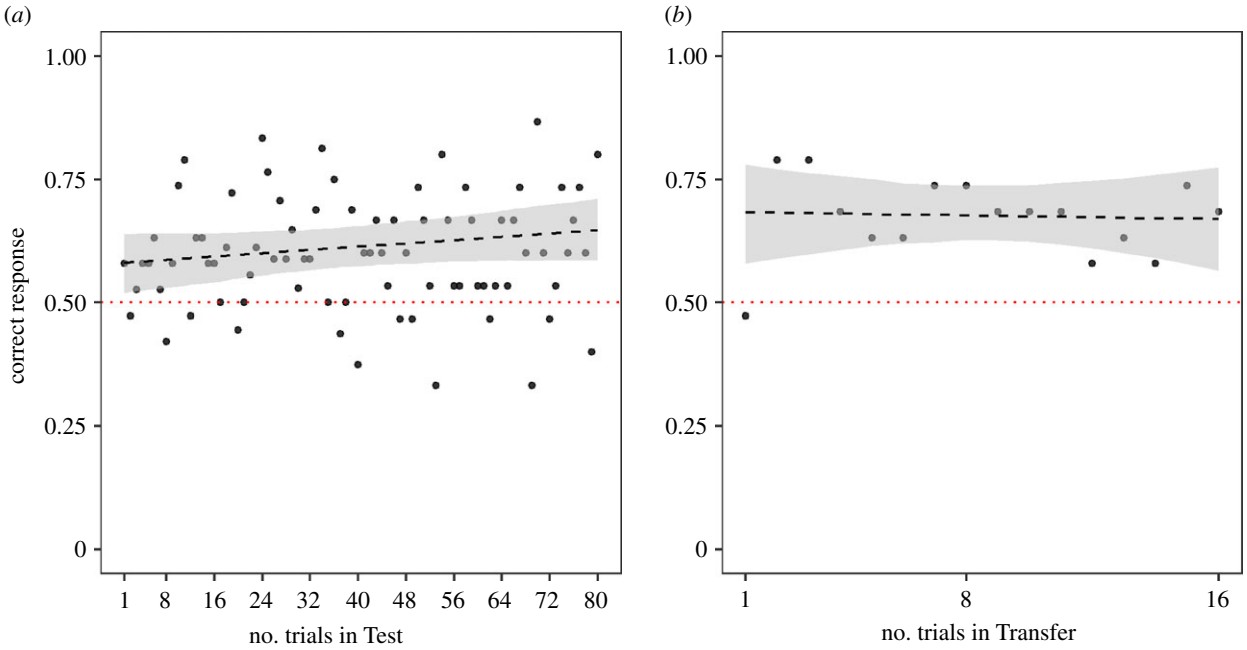

**Figure 3.** Performance of monkeys in the Test (*a*) and Transfer (*b*) phases across trials in Experiment 2. Dotted line shows the chance level. Dashed line shows the model predictions (models were fitted separately for each phase; all conducted trials were included in these models) and the shaded area is the bootstrapped 95% confidence interval. (Online version in colour.)

of 10 (80 trials). Most of the monkeys (15 of 19) completed the maximum number of sessions before they moved on to the Transfer phase without meeting the criterion (electronic supplementary material, table S7). The binomial tests for each of the 15 'unsuccessful' individuals showed that when considering all 80 trials in the Test phase, five of them performed significantly above chance.

When comparing the last 16 trials of the Test and the 16 trials of the Transfer phases, the monkeys performed significantly above chance level in both Test (Est. ± s.e.: 0.78 ± 0.17, $z = 4.63$, $p < 0.001$) and Transfer (0.72 ± 0.13, $z = 5.61$, $p < 0.001$) (figure 3). They also performed significantly above chance in both trial types (food–stick: 0.97 ± 0.17, $z = 5.84$, $p < 0.001$ and stick–food: 0.53 ± 0.12, $z = 4.34$, $p < 0.001$).

The full model comprising of phase, trial type and their interaction as well as age, sex and trial number fit the data better than the null model which lacked these fixed effects ($\chi_5^2 = 13.49$, $p < 0.05$, electronic supplementary material, table S8). However, the interaction term was not significant and was removed. The reduced model also fitted the data better than the null model ($\chi_4^2 = 13.58$, $p < 0.01$, electronic supplementary material, table S9). Monkeys performed better in the food–stick trials than stick–food trials ($\chi_1^2 = 5.22$, $p < 0.05$) and males performed better than females ($\chi_1^2 = 7.59$, $p < 0.01$). The sample was unbalanced with regards to sex due to availability sampling; therefore, sex differences were not analysed further.

There was no effect of phase ($\chi_1^2 = 0.06$, $p = 0.808$), age ($\chi_1^2 = 1.59$, $p = 0.207$) nor trial number ($\chi_1^2 = 0.02$, $p = 0.890$). These results indicated that monkeys' overall performance was not influenced by the change in the order of events in the Transfer phase. However, the first trial analysis of the Transfer phase showed that monkeys performed at chance level (9/19 individuals chose the correct cup, binomial test: $p = 0.18$).

We further found that the Transfer performance of the monkeys that reached the criterion in the Test phase ($N = 4$) did not differ from those that did not reach the criterion in the Test phase ($N = 15$) ($\chi_2^2 = 5.45$, $p = 0.065$, electronic supplementary material, table S10).

## (d) Discussion

When we analysed the first eight trials of the Test phase (the number of children received), monkeys' performance did not deviate from chance level. However, over a greater number of trials, they located the reward in the correct cup at above chance levels, with some reaching criterion in the Test phase. Performance over the 16 trials of the Transfer phase showed that they could adapt to the change in the order of events. However, the first trial analysis of the Transfer showed that, like younger children, monkeys did not spontaneously adjust their choice to the order of events.

We found a significant effect of trial type (better performance in the food–stick trials than in stick–food trials). Differing attention and memory demands associated with these trial types may be a plausible explanation for this finding. In the food–stick trials, once the subject saw one cup was ripped after the food was picked up, it did not need to pay attention to the rest of the trial to solve the task, provided it could remember the location of the first ripped foil. Indeed, we observed that in food–stick trials, after the first cup was revealed to be ripped (by the food), the monkeys would often move to that side of the cubicle and reach through the hole in the window until the end of the trial without paying any attention to E's next actions (i.e. picking up the stick to rip the other cup). On the other hand, in the stick-food trials, they had to remember which cup was ripped initially to locate the reward in the correct cup after the food hiding event. Paying attention only to the result of the food hiding event would not lead to success because by then, both foils are ripped. The monkeys performed above chance in both trial types, therefore better performance in one would not explain their overall success. However, it might point to the role of executive functions (i.e. working

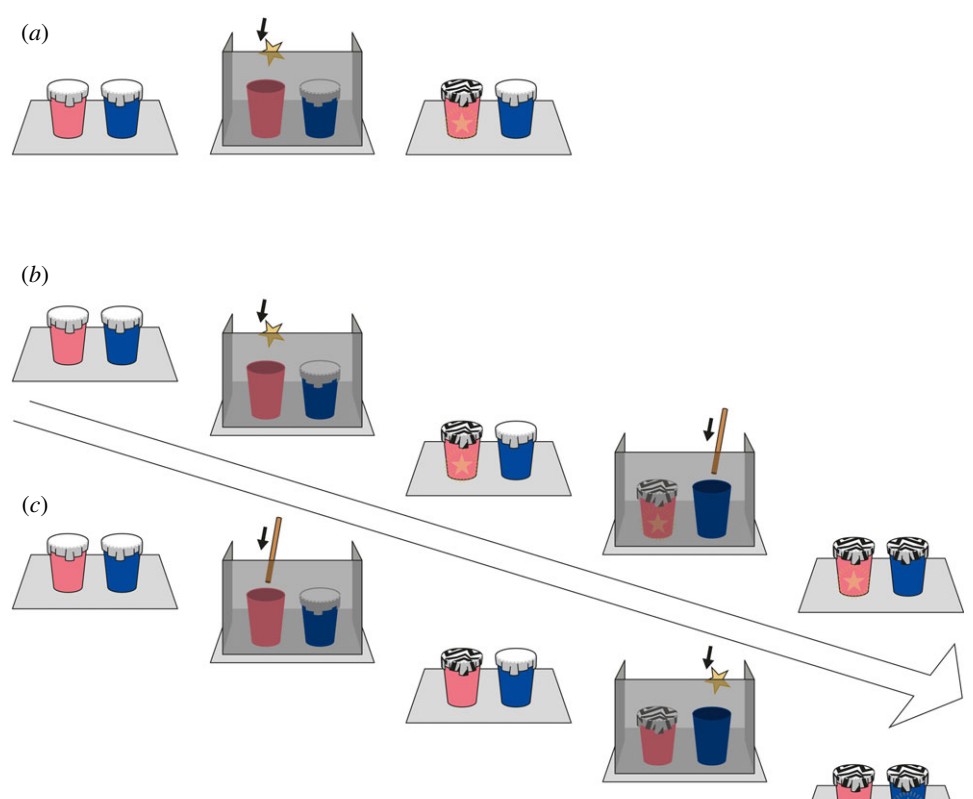

**Figure 4.** A visual illustration of (*a*) Pre-test 2, (*b*) Test and (*c*) Transfer phases of Experiment 3. The illustration shows a food–stick trial for the Test phase and a stick–food trial for the Transfer phase (the order of the conditions was counterbalanced across participants). The star represents the food-reward. The diagonal arrow shows the temporal progression of the trial. (Online version in colour.)

memory and attentional flexibility) in problem-solving based on indirect visual cues.

Overall, the monkeys received a larger number of warm-up, pre-test and test trials than the children. We carried out these additional phases and multiple trials to ensure that the monkeys could use the ripped foil as a cue to locate the reward and keep track of two events before they moved on to the critical Transfer phase. However, given this extensive experience, the monkeys might have learnt a rather complex arbitrary rule: 'Focus on the state change after E picks up the reward'. We address this possibility in Experiment 3.

## 4. Experiment 3—arbitrary follow-up with capuchin monkeys

Instead of covering the cups with foil as in Experiments 1&2, in this experiment, E covered them with white paper, and instead of the state change following the event behind barrier being that one of the foil covers was ripped, it was that one by one these white covers were exchanged for covers painted with black zigzags. If the monkeys represented the causal relationship between the state change in foil and the food's hiding, we expected weaker performance when the cues were arbitrarily related to the location of the reward.

### (a) Methods
#### (i) Participants
Fourteen capuchin monkeys of the same sample participated in this experiment ($N_{females} = 5$, $M_{age} = 9.07$, s.d.$_{age} = 3.46$) (electronic supplementary material, table S6).

### (b) Materials
The same cups were used. Instead of foil, we used two pieces of plain white paper and two pieces of white paper painted with a black zigzag to cover the cups (electronic supplementary material, figure S3).

#### (i) Procedure
Same as in Experiment 2 except for Pre-test 1.

*Pre-test 1.* E showed two empty cups to the subject and behind a barrier put a raisin in one of the cups and covered it with a black-zigzagged lid. In the end, the subject saw a cup without a lid and one covered with paper. The aim of this phase was to familiarize the monkeys with the novel lid and finding a reward in one of the cups using the zig-zagged paper as a cue. The number of sessions needed to pass this phase ranged from two to 11 ($M = 4.29$, s.d. = 2.55).

*Pre-test 2.* The two cups were covered with white paper in the beginning of the trial. In the end, the empty cup remained covered with white paper, while the cover of the baited cup was exchanged with a zigzag-patterned paper lid. All participants passed this phase within two sessions, the minimum number of sessions required.

*Test.* The same as in Experiment 2 with the only difference being that the paper lids changed from white to patterned following the events behind the barrier instead of foil.

*Transfer.* The participants received two sessions in the reverse order to the Test phase (figure 4).

#### (ii) Scoring and analysis
The same as Experiments 1&2. There were no issues regarding the model stability or multicollinearity.

## (c) Results

The number of sessions needed to pass the Test phase ranged from two to 10 ($M = 8.35$, s.d. = 3.27). Most of the monkeys (11 of 14) completed 10 sessions without meeting the criterion before they moved on to the Transfer phase.

Overall, monkeys' performance did not deviate from chance in the Test (Est. ± s.e.: $0.48 ± 0.24$, $z = 1.99$, $p = 0.139$) or in Transfer phases ($0.27 ± 0.16$, $z = 1.74$, $p = 0.162$) (electronic supplementary material, figure S4). A GLMM comprising of phase, trial type and their interaction as well as age, sex and trial number was a better fit to the data than the null model ($\chi^2_5 = 11.67$, $p < 0.05$) (electronic supplementary material, table S11). However, the interaction term was not significant ($\chi^2_1 = 0.88$, $p = 0.348$) so we removed it. The reduced model was also a better fit than the null model ($\chi^2_4 = 10.79$, $p < 0.05$, electronic supplementary material, table S12). Only trial type had a significant effect on performance ($\chi^2_1 = 5.81$, $p < 0.05$): monkeys performed significantly better than chance in food–stick trials ($0.79 ± 0.30$, $z = 2.68$, $p < 0.05$) but not in stick–food trials ($0.04 ± 0.13$, $z = 0.27$, $p = 0.789$) (electronic supplementary material, figure S5). The first trial performance in the Transfer phase did not differ from chance level (five of 14 individuals chose the correct cup, binomial test: $p = 0.424$).

## (d) Discussion

The monkeys did not perform significantly above chance levels when the cues were arbitrarily related to the location of the reward, in contrast with their performance in the causal task. These findings are consistent with the notion that the monkeys benefited from a representation of the physical object–object interactions involved in the foil task. On the other hand, in both Experiments 2 and 3, monkeys benefitted from trials that had lower demands on working memory (i.e. food–stick trials).

## 5. General discussion

Our results suggest that by 4 years of age children could make causal inferences based on visible traces, in the absence of a verbal prompt to identify the cause of an outcome. By contrast, younger preschoolers and capuchin monkeys did not show robust evidence for this ability, as they performed at chance in the first trial of transfer task. However, given additional experience capuchin monkeys were able to transfer their knowledge to the reversed order of events only when the visual traces were causally related to the food's location (Experiment 2) as opposed to when they were arbitrarily related (Experiment 3). This suggests that causal representations may play a role in their problem-solving.

Similar age differences were found in previous studies [18,22], in which 3-year olds reverted to biased responses when they were required to reason from effects to causes. This developmental difference was also found in studies where young children had to reason about hidden properties of objects such internal properties or weight [50–52], or about objects whose efficacy they never observed before [53]. Taken together, these findings suggest that the ability to reason from evidence to causes undergoes developmental change between 3 and 4 years of age. As causal reasoning is not a unitary ability but rather an emergent capacity resulting from the interplay of distinct abilities (e.g. statistical learning, domain-general processing capacity, domain-specific knowledge of causal relations, cultural input) that develop significantly during preschool years, several possibilities could explain this difference.

One possible explanation is that 3-year olds lacked the required background knowledge about the specific events and materials involved. However, we do not think this is a likely explanation. Children, as well as capuchin monkeys would all have some first-hand experience with ripping paper, cardboard boxes and leaves/grass that would provide them with the necessary prior experience to formulate this object knowledge, if they were capable of doing so. However, if young children lacked this knowledge, it would have been challenging to pass the task within the eight trials they were given. To explore this further, 3-year olds could be provided with additional experience (e.g. poking tissues or paper with sticks) to see if they generalized this knowledge to the experimental context.

Alternatively, the performance of younger children (and capuchin monkeys) on this task could be influenced by the limitations in their processing capacity. Executive functions develop significantly between 3 and 4 years [54,55]. It has been found that children's working memory capacity was a better predictor than their chronological age for their performance on reasoning tasks [56]. In this study, we did not assess children's ability to keep track of two events in the absence of inference-making requirement (but for evidence that they can do so, see [57]). Measuring causal reasoning abilities alongside executive functions would be informative to explore this further. These abilities might constrain causal reasoning, or they might mask its expression if task demands overwhelm executive functions.

Another possibility is the emergence of a more abstract notion of a causal mechanism. Children between the ages 3 and 4 years are known to have a developing knowledge about how events are related to their outcomes [13,15,16]. It is likely that their understanding is limited to certain mechanisms they have direct experience with (such as toys working with batteries) and over time they begin to generalize this knowledge and develop an explicit understanding that all causes lead to their effects through a mechanism. One route to doing this might be through the use of language. Especially in cases where children do not have access to the causal information visually, verbal input from others (e.g. instructions, explicit teaching, alerting to the presence of a mechanism, requests for explanations) becomes a valuable source of information [9]. Once children learn that the causes of certain events may not be directly observable, they may form a new concept, 'hidden causes', which may act as a placeholder for further learning [10]. This would not necessarily specify the cause of a particular event, but it could motivate children to look for causes when faced with indirect evidence.

This explanation is consistent with a previous study in which 4-year olds (but not 3-year olds) used indirect auditory cues to locate a fallen object [25]. In that study, we ran a set of follow-up experiments to explore the reasons for 3-year-olds' failure and found that adding causal framing to the test question improved 3-year-olds' performance. Further research could examine if in this task, changing the question from 'where is the sticker?' to 'which cover did I rip with the sticker?' would boost performance in this age group. This kind of causal framing might provide an important scaffold for children's emerging ability to identify and diagnose

causes from indirect information, both in an experimental context and in the real world.

While the performance of the monkeys on the initial trials of this task were similar to the younger preschoolers, with additional trials capuchins learnt to locate the reward in the correct cup, and they were able to flexibly transfer this knowledge to a variant of the task where the order of events reversed. This successful performance when there were real physical causal relations as opposed to their chance-level performance in the arbitrary follow-up suggests that representation of causal object–object interactions played a role in their ability to learn to use physical traces to locate rewards. This is in line with a previous study where monkeys successfully used causal (e.g. shaken baited cup) but not arbitrary auditory cues (e.g. recorded sound of a shaken cup) to locate a reward [36,58]. The additional exposure might have facilitated performance by making them more familiar with the causal relations involved. Research using different paradigms has shown that the performance of both capuchin monkeys and chimpanzees in novel tasks improved after repeated practice [59–61], especially if they received the task in progressively increasing levels of difficulty [59]. It might be that, in the absence of direct visual experience of causal mechanisms, non-human primates need more experience to 'fill in the gap'. Future work could explore whether 3-year olds would show a similar competence.

To conclude, we introduced a novel paradigm that made it possible to explore inference making without verbal prompts to look for and identify causes in children and non-human primates. Our results suggest that by 4 years of age, preschool children can detect unseen causes based on evidence alone. With more experience, capuchin monkeys were able to use information about physical object–object interactions to solve problems, even though these were not directly observed.

Ethics. The experiment involving children was ethically approved by University of St Andrews Teaching and Research Ethics Committee and informed consent was taken from parents/guardians. The experiments involving capuchin monkeys was conducted in accordance with the regulations of the University of St Andrews Animal Welfare and Ethics Committee and Edinburgh Zoo.

Data accessibility. The data and the R code for all the analyses can be found at https://github.com/zcivelek/Ripped-Foil-for-Proceedings-B.
The data are provided in the electronic supplementary material [62].

Authors' contributions. Z.C.: conceptualization, formal analysis, methodology, visualization, writing-original draft, writing-review and editing; C.J.V.: conceptualization, formal analysis, methodology, supervision, visualization, writing-review and editing; A.S.: conceptualization, formal analysis, funding acquisition, methodology, project administration, resources, supervision, visualization, writing-review and editing

All authors gave final approval for publication and agreed to be held accountable for the work performed therein.

Competing interests. We declare we have no competing interests

Funding. This project has received funding from the European Research Council (ERC) under the European Union's Horizon 2020 Research and Innovation Programme (grant agreement no. 639072). Edinburgh Zoo's Living Links Research Facility is core supported by the Royal Zoological Society of Scotland (registered charity no.: SC004064) through funding generated by its visitors, members and supporters.

Acknowledgements. We are grateful to the Royal Zoological Society of Scotland (RZSS) and the University of St Andrews for core financial support to the RZSS Edinburgh Zoo's Living Links Research Facility where this project was carried out. We are grateful to the RZSS keeping and veterinary staff for their care of animals and technical support throughout this project. We also thank Joao Pedro Sobrinho Ramos and Maham Haq for reliability coding, Dundee Science Centre and nurseries in St Andrews, Dundee and Cupar for letting us run our tests with children.

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
