## [Peer Review File · Proceedings of the Royal Society B: Biological Sciences]

Review History

RSPB-2020-2611.R0 (Original submission)

Review form: Reviewer 1

Recommendation

Major revision is needed (please make suggestions in comments)

Scientific importance: Is the manuscript an original and important contribution to its field?

Good

General interest: Is the paper of sufficient general interest?

Good

Quality of the paper: Is the overall quality of the paper suitable?

Acceptable

Is the length of the paper justified?

Yes

Should the paper be seen by a specialist statistical reviewer?

No

Do you have any concerns about statistical analyses in this paper? If so, please specify them explicitly in your report.

No

It is a condition of publication that authors make their supporting data, code and materials available - either as supplementary material or hosted in an external repository. Please rate, if applicable, the supporting data on the following criteria.

Is it accessible?

N/A

Is it clear?

N/A

Is it adequate?

N/A

Do you have any ethical concerns with this paper?

No

Comments to the Author

The manuscript reports three experiments – one on preschoolers and two on capuchin monkeys, examining their capacity to make a causal inference from visual mechanistic information that unfolds over time. Overall, I have positive inclinations towards this manuscript, but I do have some concerns that I would like to see the authors address.

1) I am not sure I understand the procedure, as it is never clearly explained. Here's what I think happens: Participants see two cups covered in foil. One cup's foil is ripped, and that cup is baited with a reward. The cups are then hidden behind a screen, and a pen is used to rip the other cups foil, so now both cups have ripped foil. Participants are then asked to retrieve the interesting object. Four-year-olds can do so; three-year-olds are at chance (Exp 1). Capuchins can also do so (Exp 2), but they cannot do this when the relation between the movement of the implement that was used to rip the foil and the causal change (the zigzags on the cup in exp 3) is arbitrary. My concern is that even though I have read this manuscript many times, I'm not 100% certain that this is the procedure, as the description of the procedure keeps changing, and is often incomplete (e.g., in the abstract, it's not clear that the baiting and the ripping aren't done (or can't be done) on the same cup. A concrete recommendation is that the authors slow down and write the procedure so clearly that there is no possible way a reader can be confused. A figure detailing the procedure would also help (Figure 1 is insufficient).

This concern is particularly important because I do wonder if there are associative cues that could explain these behaviors. The arbitrary experiment (Exp 3) suggests this is not the case, but it's also possible that because the capuchins need to forage to survive, the use of this kind of temporal cue has been ramped up, much like the Garcia effect in rats. Put another way, the authors conclude that this causal reasoning capacity is shared between human beings and non-human primates, but it is also possible that what the monkeys have evolved is a domain-specific associative strategy that allows them to solve this particular task (one that involves foraging), but not causal inferences from temporal information more generally? This strikes me as a clear alternative to the authors' interpretation, which needs to be addressed.

2) Assuming that I have at least some aspect of the procedure correct, the task reminds me a great deal of work by Teresa McCormack and Christoph Hoerl on causal reasoning through temporal sequencing (e.g., McCormack & Hoerl, 2005). It seems to me that the procedure used there is very similar to the procedure used here. They find that it isn't until age 5 that children succeed (they did not test non-human primates to my knowledge). It would be worthwhile for the authors to

integrate the present findings with these. Related to the point that I made in #1, would the monkeys perform similarly to the older children or the younger children in this task?

3) How was the sample size for the child sample determined? An a priori power analysis was not conducted, and no stopping rule was in place; in general, effect sizes are not reported, except in the figures. But with the figures small, this information is hard to decipher. The figures are also in black and white, so it is hard to tell the color coding (e.g., they refer to a red line, but I have no idea which line they are referring to, nor do I know what the vertical arrows represent).

4) In the developmental literature, there are several possible ways to interpret the age-related change. The authors discuss some of them (e.g., cognitive control and demand characteristics of the procedure). I agree that demand characteristics are not a strong consideration, and that cognitive control might affect performance, but in many of their citations, there is a third alternative, which is that between the ages of 3-4, children develop an explicit concept that causal relations have mechanisms. That is, they develop an explicit domain-general capacity that posits how events relate to one another. This could be further discussed, particularly in light of whether such a capacity is shared by non-human primates.

To conclude, this is an interesting manuscript. I would like to see more clarity in the methods and argumentation to be certain that there are not alternate explanations for the phenomenon. Given the difficulty I had in figuring out the exact nature of the procedure, I would like to also see a better integration with some of the existing research that I mentioned in this review.

Review form: Reviewer 2

Recommendation

Major revision is needed (please make suggestions in comments)

Scientific importance: Is the manuscript an original and important contribution to its field?

Excellent

General interest: Is the paper of sufficient general interest?

Good

Quality of the paper: Is the overall quality of the paper suitable?

Good

Is the length of the paper justified?

Yes

Should the paper be seen by a specialist statistical reviewer?

Yes

Do you have any concerns about statistical analyses in this paper? If so, please specify them explicitly in your report.

Yes

It is a condition of publication that authors make their supporting data, code and materials available - either as supplementary material or hosted in an external repository. Please rate, if applicable, the supporting data on the following criteria.

Is it accessible?

Yes

Is it clear?

Yes

Is it adequate?

Yes

Do you have any ethical concerns with this paper?

No

Comments to the Author

This is a thorough, interesting, and timely paper on the use of visual information by human children and Capuchins to find rewards after hidden interactions between objects. The clever design rules out various confounds by having the final state of the hiding locations look identical when revealed to participants, meaning they must use information previously acquired to pass. Similarly, the “transfer” concept is an effective way of ruling out simple arbitrary rule accounts of the behavioral patterns. I would also like to commend the authors for their efforts to keep the task minimalist with similarly low verbal instruction load between the children and the capuchins, and for making their data and code available online – which I found helpful in writing this review. I do have a number of questions and some concerns that I believe need to be addressed before this paper can be published, which I detail below:

Introduction. A point of clarification: is it correct to say about Capuchin feeding activity that “These are costly activities in terms of energy, and animals should engage in them only when it is highly likely that they will get a return”? The idea that the Capuchins should ONLY engage in seeking out food when returns are highly likely seems counterintuitive – is there a citation for this, or perhaps does the claim need to be tempered somewhat?

Methods.

Could the authors please clarify the following points. Was the study pre-registered? How was sample size determined in each study? What was the stopping rule for data collection? Was a power analysis conducted? More information about these sample considerations would be helpful in the manuscript.

The various criteria for passing or failing, moving on from pre-test to test phases, and so-on felt somewhat arbitrary. Could the authors please specify whether these were all determined a-priori, or if they were changed at all after the data had been collected?

For example, why exactly was the pre-test phase so different for the Capuchins and the children? And why exactly did the authors make the following change between E1 and E2 (children vs monkeys):

“In the main analyses, we used the last 16 trials of Test phase for each monkey (as opposed to all trials) and the 16 trials (2 sessions) of the Transfer phase. Since we were interested in the potential performance change from Test to Transfer, we wanted to include the monkeys’ best performance in the Test phase.”

Wouldn’t the same logic apply in both cases?

When it comes to fitting the models, the authors specify that they “left out the correlation parameters between random intercepts and random slopes terms” – was this because the lmer model did not converge without this specification? Could the authors please clarify in text their reason for this modeling decision.

Similarly, in the GLM 01 in Experiment 1 and elsewhere (experiment 3) the notes in the R script says there is a singular fit warning for some models. If there was a singular fit for given models,

did the authors attempt to rectify this issue and if so, could they please explain how? If they believe the singular fits for the lmer mixed effects models are not a problem, could they please clarify.

Discussion

I do not follow the authors explanation for the fact that the monkeys performed better in the food-stick condition. The authors write: “differing memory demands associated with these trial types seem to be a plausible explanation” and then mention executive functions – but unless I am missing something they do not elaborate, and I cannot figure out why this would be a plausible explanation for the difference.

The conclusion states: “without the need for verbal instruction in children and nonhuman primates” but this may need to be slightly re-worded as there was of course some verbal instruction for the children.

Review form: Reviewer 3

Recommendation

Accept with minor revision (please list in comments)

Scientific importance: Is the manuscript an original and important contribution to its field?

Excellent

General interest: Is the paper of sufficient general interest?

Excellent

Quality of the paper: Is the overall quality of the paper suitable?

Excellent

Is the length of the paper justified?

Yes

Should the paper be seen by a specialist statistical reviewer?

No

Do you have any concerns about statistical analyses in this paper? If so, please specify them explicitly in your report.

No

It is a condition of publication that authors make their supporting data, code and materials available - either as supplementary material or hosted in an external repository. Please rate, if applicable, the supporting data on the following criteria.

Is it accessible?

Yes

Is it clear?

Yes

Is it adequate?

Yes

Do you have any ethical concerns with this paper?

No

Comments to the Author

This paper reports an interesting investigation of human children's and capuchin monkeys' physical reasoning abilities to infer causes from visual effects that extends previous research by using a novel paradigm. The study design allows to differentiate between participants understanding of physical causes or participants using arbitrary association rules to choose between reward locations. The authors find that 4-year olds but not 3-year olds solved the experimental task successfully and they performed above chance already in their first trials. With increasing age, children also became better at reporting correct explanations for the events during the experimental procedure. On the group level, capuchin monkeys performed above chance when considering all trials. They had more difficulty to solve the task with arbitrary cause-effect relations (Exp. 3) than with causally meaningful cause-effect relations (Exp. 2).

Overall, the submission is in very good shape and was a pleasure to read. I found the manuscript easy to follow, the study is well embedded in existing relevant literature, supplementary material is appropriately referenced in the main body, and data & analysis code are available.

Experimental procedure and statistical approach are in principle suited to test the hypotheses, but I have some comments regarding the relationship of individual pass/fail performance and assessment of group-level performance in the monkey experiments (see below). This issue requires clarification (either when introducing the success criterion or in the discussion or both) or a more consistent data assessment approach. I only have a few other comments, which I am confident the authors will be able to address.

ESM

- ESM 34-37 Why was the effect of age assessed in a separate model comparison?
- ESM Table S2: Why do you provide χ^2 and df values for some but not other terms (applies to this and other tables)? I found the combination of table notation and footnote a bit hard to read and recommend to only name the term (e.g. "Trial type") bar the reported category ("sticker-pen") in the table rows. Especially the rows with interaction terms would become easier to read. By providing information on reference categories in the note (as you already do) all information is available.

Experiment 2

- L 250-255: Where there any monkeys who didn't pass the pre-test?
- L 243-259 The "pass" criterion is not fully clear to me both in terms of its application and in terms of its pass/fail consequences. Could "14 of 16 trials correct" be reached across three sessions or did the monkeys have to perform at this level in two consecutive sessions? It also appears that no monkey was ever excluded from proceeding to the next stage when failing to reach the criterion?
- It came as a surprise to me that on the one hand, 15 of 19 individuals didn't reach the criterion in the Test condition at all, but on the other hand group performance was above chance. Doesn't this mean the set criterion is not a good indicator of an individual's task performance? I find it a bit odd that on the one hand, individuals are categorized as having "passed" a pre-defined criterion and are not further tested in this test phase (they move on to the transfer phase immediately), while on the other hand other individuals have not passed the criterion but are not categorized as "failed"; instead their data of 80 trials is used to analyse group performance whereas we don't know how the "successful" individuals would have performed in the remaining sessions. Speaking from own experience, I repeatedly encountered that passing a numerical criterion doesn't mean that a monkey keeps performing at above chance levels if tested further. The authors' approach thus seems a bit inconsistent here and requires more explanation and discussion of function and consequences of the pre-defined criterion and how it relates to the reported group performance.

- Also, l 302-308 should be moved further up (around l 279-282) so the reader immediately learns that most individuals failed to reach the criterion despite above chance group level performance.
- It would be interesting to learn if performance of “successful” individuals differs from that of “failed” individuals in the following transfer phase.
- To complement performance assessment of all trials for the whole group, the authors might also want to report how many of the 15 individuals who never reached the numerical 14 of 16 criterion performed above chance when all their trials were considered.
- L 307-308: How can a monkey reach a criterion of 14 of 16 correct choices (as described in l 256/257) after only 8 trials?

Experiment 3

- L 333: Typo -> delete “if they”

Decision letter (RSPB-2020-2611.R0)

12-Jan-2021

Dear Dr Civelek:

I am with regard to your manuscript RSPB-2020-2611 entitled "What happened? Do preschool children and capuchin monkeys spontaneously use visual traces to locate a reward?" Your manuscript has been seen by three reviewers and the Associate Editor, and while all of them see a lot of potential in your paper, they note a number of issues. In particular, they identify several places where the methodology and statistics are difficult for a general reader to follow, and they and the AE ask that you ground your paper more firmly in the relevant theoretical framework. I second the AE's point that while your paper is appropriately framed for an animal cognition journal, I think that you could make a much stronger impact across a broader readership by emphasizing the theoretical importance of the combined comparative/developmental approach in general, and your work specifically. Thus, I am rejecting your paper in its current form, but will be happy to consider a resubmission, provided the comments of the referees are fully addressed. However please note that this is not a provisional acceptance.

Sincerely,
Dr Sarah Brosnan
Editor, Proceedings B
mailto: proceedingsb@royalsociety.org

Associate Editor

Board Member: 1

Comments to Author:

This manuscript examines understanding of causality in children over development and in capuchin monkeys. Understanding of causality has emerged as an important cognitive skill with repercussions for phenomena as far ranging from animal foraging to human capacity to engage in logical or scientific thought, and this set of studies develop a clever way to address its developmental and phylogenetic roots. All three reviewers agree this is an interesting topic, and have helpful comments regarding the paper that should be addressed in revision.

One crucial point raised in the reviews is to make sure the procedure is understandable by a general readership. The reviewers raise several points about the procedure and statistical approach that should be either clarified or clearly addressed as relevant. This includes ensuring that the actual procedure is comprehensible by a general audience; addressing several questions about the statistical analyses; and clarifying decisions about passing criteria (amongst others). Along these same lines, R1 and R2 raise some potential alternative interpretations that, while not necessarily radically altering the paper's outcome, should be addressed to allow for a more nuanced discussion of the implications of the results.

A final major point is to provide a more robust theoretical framework for both the child and monkey results. R1 provides some references and other possible interpretations (for example, of the age effect) that could be helpful to this end. It is also necessary to pay close attention to justifying the need for developmental and comparative work on this topic in the introduction. For example, what are the repercussions for proposals about uniquely-human cognition and/or behaviour if capuchins do or do not have this capacity? Why is it necessary to study children? At the moment, the paper is framed in a way that is more appropriate for an animal cognition or development journal, but the stakes here need to be made apparent to a general readers.

Reviewer(s)' Comments to Author:

Referee: 1

Comments to the Author(s)

The manuscript reports three experiments – one on preschoolers and two on capuchin monkeys, examining their capacity to make a causal inference from visual mechanistic information that unfolds over time. Overall, I have positive inclinations towards this manuscript, but I do have some concerns that I would like to see the authors address.

1) I am not sure I understand the procedure, as it is never clearly explained. Here's what I think happens: Participants see two cups covered in foil. One cup's foil is ripped, and that cup is baited with a reward. The cups are then hidden behind a screen, and a pen is used to rip the other cups foil, so now both cups have ripped foil. Participants are then asked to retrieve the interesting object. Four-year-olds can do so; three-year-olds are at chance (Exp 1). Capuchins can also do so (Exp 2), but they cannot do this when the relation between the movement of the implement that was used to rip the foil and the causal change (the zigzags on the cup in exp 3) is arbitrary. My concern is that even though I have read this manuscript many times, I'm not 100% certain that this is the procedure, as the description of the procedure keeps changing, and is often incomplete

(e.g., in the abstract, it's not clear that the baiting and the ripping aren't done (or can't be done) on the same cup. A concrete recommendation is that the authors slow down and write the procedure so clearly that there is no possible way a reader can be confused. A figure detailing the procedure would also help (Figure 1 is insufficient).

This concern is particularly important because I do wonder if there are associative cues that could explain these behaviors. The arbitrary experiment (Exp 3) suggests this is not the case, but it's also possible that because the capuchins need to forage to survive, the use of this kind of temporal cue has been ramped up, much like the Garcia effect in rats. Put another way, the authors conclude that this causal reasoning capacity is shared between human beings and non-human primates, but it is also possible that what the monkeys have evolved is a domain-specific associative strategy that allows them to solve this particular task (one that involves foraging), but not causal inferences from temporal information more generally? This strikes me as a clear alternative to the authors' interpretation, which needs to be addressed.

2) Assuming that I have at least some aspect of the procedure correct, the task reminds me a great deal of work by Teresa McCormack and Christoph Hoerl on causal reasoning through temporal sequencing (e.g., McCormack & Hoerl, 2005). It seems to me that the procedure used there is very similar to the procedure used here. They find that it isn't until age 5 that children succeed (they did not test non-human primates to my knowledge). It would be worthwhile for the authors to integrate the present findings with these. Related to the point that I made in #1, would the monkeys perform similarly to the older children or the younger children in this task?

3) How was the sample size for the child sample determined? An a priori power analysis was not conducted, and no stopping rule was in place; in general, effect sizes are not reported, except in the figures. But with the figures small, this information is hard to decipher. The figures are also in black and white, so it is hard to tell the color coding (e.g., they refer to a red line, but I have no idea which line they are referring to, nor do I know what the vertical arrows represent).

4) In the developmental literature, there are several possible ways to interpret the age-related change. The authors discuss some of them (e.g., cognitive control and demand characteristics of the procedure). I agree that demand characteristics are not a strong consideration, and that cognitive control might affect performance, but in many of their citations, there is a third alternative, which is that between the ages of 3-4, children develop an explicit concept that causal relations have mechanisms. That is, they develop an explicit domain-general capacity that posits how events relate to one another. This could be further discussed, particularly in light of whether such a capacity is shared by non-human primates.

To conclude, this is an interesting manuscript. I would like to see more clarity in the methods and argumentation to be certain that there are not alternate explanations for the phenomenon. Given the difficulty I had in figuring out the exact nature of the procedure, I would like to also see a better integration with some of the existing research that I mentioned in this review.

Referee: 2

Comments to the Author(s)

This is a thorough, interesting, and timely paper on the use of visual information by human children and Capuchins to find rewards after hidden interactions between objects. The clever design rules out various confounds by having the final state of the hiding locations look identical when revealed to participants, meaning they must use information previously acquired to pass. Similarly, the "transfer" concept is an effective way of ruling out simple arbitrary rule accounts of the behavioral patterns. I would also like to commend the authors for their efforts to keep the task minimalist with similarly low verbal instruction load between the children and the capuchins, and for making their data and code available online - which I found helpful in writing this review. I do have a number of questions and some concerns that I believe need to be addressed before this paper can be published, which I detail below:

Introduction. A point of clarification: is it correct to say about Capuchin feeding activity that “These are costly activities in terms of energy, and animals should engage in them only when it is highly likely that they will get a return”? The idea that the Capuchins should ONLY engage in seeking out food when returns are highly likely seems counterintuitive – is there a citation for this, or perhaps does the claim need to be tempered somewhat?

Methods.

Could the authors please clarify the following points. Was the study pre-registered? How was sample size determined in each study? What was the stopping rule for data collection? Was a power analysis conducted? More information about these sample considerations would be helpful in the manuscript.

The various criteria for passing or failing, moving on from pre-test to test phases, and so-on felt somewhat arbitrary. Could the authors please specify whether these were all determined a-priori, or if they were changed at all after the data had been collected?

For example, why exactly was the pre-test phase so different for the Capuchins and the children? And why exactly did the authors make the following change between E1 and E2 (children vs monkeys):

“In the main analyses, we used the last 16 trials of Test phase for each monkey (as opposed to all trials) and the 16 trials (2 sessions) of the Transfer phase. Since we were interested in the potential performance change from Test to Transfer, we wanted to include the monkeys’ best performance in the Test phase.”

Wouldn’t the same logic apply in both cases?

When it comes to fitting the models, the authors specify that they “left out the correlation parameters between random intercepts and random slopes terms” – was this because the lmer model did not converge without this specification? Could the authors please clarify in text their reason for this modeling decision.

Similarly, in the GLM 01 in Experiment 1 and elsewhere (experiment 3) the notes in the R script says there is a singular fit warning for some models. If there was a singular fit for given models, did the authors attempt to rectify this issue and if so, could they please explain how? If they believe the singular fits for the lmer mixed effects models are not a problem, could they please clarify.

Discussion

I do not follow the authors explanation for the fact that the monkeys performed better in the food-stick condition. The authors write: “differing memory demands associated with these trial types seem to be a plausible explanation” and then mention executive functions – but unless I am missing something they do not elaborate, and I cannot figure out why this would be a plausible explanation for the difference.

The conclusion states: “without the need for verbal instruction in children and nonhuman primates” but this may need to be slightly re-worded as there was of course some verbal instruction for the children.

Referee: 3

Comments to the Author(s)

This paper reports an interesting investigation of human children’s and capuchin monkeys’ physical reasoning abilities to infer causes from visual effects that extends previous research by using a novel paradigm. The study design allows to differentiate between participants

understanding of physical causes or participants using arbitrary association rules to choose between reward locations. The authors find that 4-year olds but not 3-year olds solved the experimental task successfully and they performed above chance already in their first trials. With increasing age, children also became better at reporting correct explanations for the events during the experimental procedure. On the group level, capuchin monkeys performed above chance when considering all trials. They had more difficulty to solve the task with arbitrary cause-effect relations (Exp. 3) than with causally meaningful cause-effect relations (Exp. 2).

Overall, the submission is in very good shape and was a pleasure to read. I found the manuscript easy to follow, the study is well embedded in existing relevant literature, supplementary material is appropriately referenced in the main body, and data & analysis code are available.

Experimental procedure and statistical approach are in principle suited to test the hypotheses, but I have some comments regarding the relationship of individual pass/fail performance and assessment of group-level performance in the monkey experiments (see below). This issue requires clarification (either when introducing the success criterion or in the discussion or both) or a more consistent data assessment approach. I only have a few other comments, which I am confident the authors will be able to address.

ESM

- ESM 34-37 Why was the effect of age assessed in a separate model comparison?
- ESM Table S2: Why do you provide χ^2 and df values for some but not other terms (applies to this and other tables)? I found the combination of table notation and footnote a bit hard to read and recommend to only name the term (e.g. "Trial type") bar the reported category ("sticker-pen") in the table rows. Especially the rows with interaction terms would become easier to read. By providing information on reference categories in the note (as you already do) all information is available.

Experiment 2

- L 250-255: Where there any monkeys who didn't pass the pre-test?
- L 243-259 The "pass" criterion is not fully clear to me both in terms of its application and in terms of its pass/fail consequences. Could "14 of 16 trials correct" be reached across three sessions or did the monkeys have to perform at this level in two consecutive sessions? It also appears that no monkey was ever excluded from proceeding to the next stage when failing to reach the criterion?
 - It came as a surprise to me that on the one hand, 15 of 19 individuals didn't reach the criterion in the Test condition at all, but on the other hand group performance was above chance. Doesn't this mean the set criterion is not a good indicator of an individual's task performance? I find it a bit odd that on the one hand, individuals are categorized as having "passed" a pre-defined criterion and are not further tested in this test phase (they move on to the transfer phase immediately), while on the other hand other individuals have not passed the criterion but are not categorized as "failed"; instead their data of 80 trials is used to analyse group performance whereas we don't know how the "successful" individuals would have performed in the remaining sessions. Speaking from own experience, I repeatedly encountered that passing a numerical criterion doesn't mean that a monkey keeps performing at above chance levels if tested further. The authors' approach thus seems a bit inconsistent here and requires more explanation and discussion of function and consequences of the pre-defined criterion and how it relates to the reported group performance.
 - Also, l 302-308 should be moved further up (around l 279-282) so the reader immediately learns that most individuals failed to reach the criterion despite above chance group level performance.
 - It would be interesting to learn if performance of "successful" individuals differs from that of "failed" individuals in the following transfer phase.
 - To complement performance assessment of all trials for the whole group, the authors might also want to report how many of the 15 individuals who never reached the numerical 14 of 16 criterion performed above chance when all their trials were considered.

- L 307-308: How can a monkey reach a criterion of 14 of 16 correct choices (as described in I 256/257) after only 8 trials?

Experiment 3

- L 333: Typo -> delete "if they"

Author's Response to Decision Letter for (RSPB-2020-2611.R0)

See Appendix A.

RSPB-2021-1101.R0

Review form: Reviewer 1

Recommendation

Accept with minor revision (please list in comments)

Scientific importance: Is the manuscript an original and important contribution to its field?

Acceptable

General interest: Is the paper of sufficient general interest?

Acceptable

Quality of the paper: Is the overall quality of the paper suitable?

Good

Is the length of the paper justified?

Yes

Should the paper be seen by a specialist statistical reviewer?

No

Do you have any concerns about statistical analyses in this paper? If so, please specify them explicitly in your report.

No

It is a condition of publication that authors make their supporting data, code and materials available - either as supplementary material or hosted in an external repository. Please rate, if applicable, the supporting data on the following criteria.

Is it accessible?

Yes

Is it clear?

No

Is it adequate?

No

Do you have any ethical concerns with this paper?

No

Comments to the Author

This is a revised version of a manuscript I reviewed previously (I was Reviewer X). My main concern previously was with the clarity of the manuscript and experiment, and the authors have done an excellent job improving the manuscript with this goal in mind. I do have some minor remaining concerns, but in general, I am highly satisfied with this revision.

1) I did not understand what was being analyzed in lines 240-242 under the phrase “open ended analysis” – a sentence or two to roadmap the reader might help here.

2a) Much in the same way that the addition of the text in Figure 1 (And the better clarity of the figure itself) facilitated an understanding of the procedure in Experiments 1-2, a figure for Experiment 3 would allow the reader to understand the distinction between Experiment 2 and 3 better. To be perfectly honest, I did not understand what was specifically different about Experiments 2 and 3, and the manuscript suffers from a lack of clarity here.

2b) I also wonder if a comparison between Experiments 2-3 is warranted, given that there seems to be some suggestion that performance on these measures should be different.

3) Finally – and I will admit that this is more a musing than a direct suggestion – I wonder to what extent children have statistical learning capacities early on, whilst their causal reasoning capacities emerge from those capacities and are limited by information processing demands, which might motivate the difference between the younger and older children in the sample. That is, if those information processing demands are reduced, or if children have more explicit knowledge to allow them to engage in retrospective reasoning, then 3-year-olds would look more like the older children in the sample (but the non-human primates would not show such a benefit, indicating a uniquely human system – although if they do, then that would speak against such an argument).

Again, this is musing, but to give concrete feedback, I would have liked to see more of a discussion of the possible mechanisms of development in the GD, particularly looking at places where children show the developmental difference between 3-4. What do those researchers suggest is developing and does that relate here? My guess is that this is more about the set-up of the experiment and the inference that children have to make about changes of state than about retrospective inferences, per se. Again, I feel like Experiment 3 can shed some insight on this idea, but because it was not as clearly presented as the other two experiments, it was more difficult to tell.

To conclude, again, this is a strong revision and a very nice piece of work.

Review form: Reviewer 2**Recommendation**

Accept as is

Scientific importance: Is the manuscript an original and important contribution to its field?

Excellent

General interest: Is the paper of sufficient general interest?

Excellent

Quality of the paper: Is the overall quality of the paper suitable?

Excellent

Is the length of the paper justified?

Yes

Should the paper be seen by a specialist statistical reviewer?

No

Do you have any concerns about statistical analyses in this paper? If so, please specify them explicitly in your report.

No

It is a condition of publication that authors make their supporting data, code and materials available - either as supplementary material or hosted in an external repository. Please rate, if applicable, the supporting data on the following criteria.

Is it accessible?

Yes

Is it clear?

Yes

Is it adequate?

Yes

Do you have any ethical concerns with this paper?

No

Comments to the Author

I believe the authors have sufficiently addressed my initial round of comments, as well as the various important issues raised by the other reviewers. I have only one minor follow-up query about the new examples in the introduction, which seem somewhat detached from the phenomenon that the experimental protocol assesses. The authors write: "(e.g., tapping on the nut to infer its fullness or weighing a tool before using it to crack a nut)" – though both of these examples involve the animal making some kind of environmental manipulation in order to gather information. In the experiments here, the animal witnesses a hidden event and is then presented with the opportunity to act upon an inferred causal relationship that took place during that hidden event. Are there naturalistic examples that more closely parallel this structure than the "tapping on the nut" or "weighing a tool" examples? Perhaps I am misunderstanding either the examples or the task itself, and if so, I would appreciate any clarification that the authors could provide.

I think this paper will make a useful and timely addition to the literature on the developmental and evolutionary roots of causal understanding. Thank you for the opportunity to review this interesting research and best wishes moving forward with this line of work.

Decision letter (RSPB-2021-1101.R0)

08-Jun-2021

Dear Dr Civelek:

Thank you for a very nice revision of your manuscript. The reviewers, Associate Editor and I all appreciate your efforts and find it much more clear. You will see that there remain a few minor issues that the reviewers and AE highlight in their comments, and I invite you to revise your manuscript to address them. We do not anticipate sending your manuscript back out for review.

Research ethics:

Use of animals and field studies:

It is a condition of publication that you make available the data and research materials supporting the results in the article (<https://royalsociety.org/journals/authors/author-guidelines/#data>). Datasets should be deposited in an appropriate publicly available repository and details of the associated accession number, link or DOI to the datasets must be included in the Data Accessibility section of the article (<https://royalsociety.org/journals/ethics-policies/data-sharing-mining/>). Reference(s) to datasets should also be included in the reference list of the article with DOIs (where available).

Please submit a copy of your revised paper within three weeks. If we do not hear from you within this time your manuscript will be rejected. If you are unable to meet this deadline please let us know as soon as possible, as we may be able to grant a short extension.

Best wishes,
Dr Sarah Brosnan
mailto:proceedingsb@royalsociety.org

Associate Editor
Comments to Author:

The reviewers and I agree that the authors have done a nice job addressing the points raised in the previous round of review, which has clarified many key aspects of the paper and strengthened the rationale of the work.

The reviewers have a few final comments clarifying aspects of the paper and its interpretation. One thing to note is the comment from R1 concerning a diagram for study 3. Along those lines, I will also note that while the current diagram is quite helpful, portions of it are extremely small and what I believe is intended to be an image of a sticker appears to be a tiny figure (only by zooming in did I see it was an animal, which is somewhat confusing). A second is the important point from R2 concerning the difference between understanding the outcome one's own manipulations on the world, versus learning about causality via observation. Given the focus in comparative work on the role of tool use in spurring the emergence of such cognitive skills, it would be worth it to mention this distinction.

I would also suggest a bit more nuance in the discussion of the capuchin results. The general discussion states that "capuchin monkeys did not show convincing evidence for this ability, as they performed at chance in the first trial of transfer task." While is true, capuchins also performed quite well overall in the transfer; their trajectory in the transfer seems different than in the original test (e.g., they do not slowly re-learn this relationship; their performance jumps on trial 2); and study 3 supports the claim that they are not just using associative learning mechanisms. In that sense, they seem to show some differences from the younger children as well.

Reviewer(s)' Comments to Author:

Referee: 1

Comments to the Author(s).

This is a revised version of a manuscript I reviewed previously (I was Reviewer X). My main concern previously was with the clarity of the manuscript and experiment, and the authors have done an excellent job improving the manuscript with this goal in mind. I do have some minor remaining concerns, but in general, I am highly satisfied with this revision.

1) I did not understand what was being analyzed in lines 240-242 under the phrase “open ended analysis” – a sentence or two to roadmap the reader might help here.

2a) Much in the same way that the addition of the text in Figure 1 (And the better clarity of the figure itself) facilitated an understanding of the procedure in Experiments 1-2, a figure for Experiment 3 would allow the reader to understand the distinction between Experiment 2 and 3 better. To be perfectly honest, I did not understand what was specifically different about Experiments 2 and 3, and the manuscript suffers from a lack of clarity here.

2b) I also wonder if a comparison between Experiments 2-3 is warranted, given that there seems to be some suggestion that performance on these measures should be different.

3) Finally – and I will admit that this is more a musing than a direct suggestion – I wonder to what extent children have statistical learning capacities early on, whilst their causal reasoning capacities emerge from those capacities and are limited by information processing demands, which might motivate the difference between the younger and older children in the sample. That is, if those information processing demands are reduced, or if children have more explicit knowledge to allow them to engage in retrospective reasoning, then 3-year-olds would look more like the older children in the sample (but the non-human primates would not show such a benefit, indicating a uniquely human system – although if they do, then that would speak against such an argument).

Again, this is musing, but to give concrete feedback, I would have liked to see more of a discussion of the possible mechanisms of development in the GD, particularly looking at places where children show the developmental difference between 3-4. What do those researchers suggest is developing and does that relate here? My guess is that this is more about the set-up of the experiment and the inference that children have to make about changes of state than about retrospective inferences, per se. Again, I feel like Experiment 3 can shed some insight on this idea, but because it was not as clearly presented as the other two experiments, it was more difficult to tell.

To conclude, again, this is a strong revision and a very nice piece of work.

Referee: 2

Comments to the Author(s).

I believe the authors have sufficiently addressed my initial round of comments, as well as the various important issues raised by the other reviewers. I have only one minor follow-up query about the new examples in the introduction, which seem somewhat detached from the phenomenon that the experimental protocol assesses. The authors write: “(e.g., tapping on the nut to infer its fullness or weighing a tool before using it to crack a nut)” – though both of these examples involve the animal making some kind of environmental manipulation in order to gather information. In the experiments here, the animal witnesses a hidden event and is then presented with the opportunity to act upon an inferred causal relationship that took place during that hidden event. Are there naturalistic examples that more closely parallel this structure than the “tapping on the nut” or “weighing a tool” examples? Perhaps I am misunderstanding either the examples or the task itself, and if so, I would appreciate any clarification that the authors could provide.

I think this paper will make a useful and timely addition to the literature on the developmental and evolutionary roots of causal understanding. Thank you for the opportunity to review this interesting research and best wishes moving forward with this line of work.

Author's Response to Decision Letter for (RSPB-2021-1101.R0)

See Appendices B & C.

Decision letter (RSPB-2021-1101.R1)

12-Jul-2021

Dear Dr Civelek

I am pleased to inform you that your manuscript entitled "What happened? Do preschool children and capuchin monkeys spontaneously use visual traces to locate a reward?" has been accepted for publication in Proceedings B.

Data Accessibility section

Open Access

Paper charges

Sincerely,

Dr Sarah Brosnan
Editor, Proceedings B
mailto: proceedingsb@royalsociety.org

Associate Editor:

Board Member

Comments to Author:

The authors have done a nice job on the revision. I specifically note that the new illustrations of the study procedures are quite helpful, and should clarify aspects of the procedure that came up in review for readers.

Appendix A

Reviewer(s)' Comments to Author (highlighted in grey) & Authors' responses to the Reviewers (below):
Line numbers are from the 'tracked changes' documents – both for the manuscript and the Electronic Supplementary Materials.

Associate Editor

Board Member: 1

Comments to Author:

This manuscript examines understanding of causality in children over development and in capuchin monkeys. Understanding of causality has emerged as an important cognitive skill with repercussions for phenomena as far ranging from animal foraging to human capacity to engage in logical or scientific thought, and this set of studies develop a clever way to address its developmental and phylogenetic roots. All three reviewers agree this is an interesting topic and have helpful comments regarding the paper that should be addressed in revision.

Thank you!

One crucial point raised in the reviews is to make sure the procedure is understandable by a general readership. The reviewers raise several points about the procedure and statistical approach that should be either clarified or clearly addressed as relevant. This includes ensuring that the actual procedure is comprehensible by a general audience; addressing several questions about the statistical analyses; and clarifying decisions about passing criteria (amongst others). Along these same lines, R1 and R2 raise some potential alternative interpretations that, while not necessarily radically altering the paper's outcome, should be addressed to allow for a more nuanced discussion of the implications of the results.

Thank you for this summary of the major issues. These are now all addressed in our responses to the Referee's comments below.

A final major point is to provide a more robust theoretical framework for both the child and monkey results. R1 provides some references and other possible interpretations (for example, of the age effect) that could be helpful to this end. It is also necessary to pay close attention to justifying the need for developmental and comparative work on this topic in the introduction. For example, what are the repercussions for proposals about uniquely-human cognition and/or behaviour if capuchins do or do not have this capacity? Why is it necessary to study children? At the moment, the paper is framed in a way that is more appropriate for an animal cognition or development journal, but the stakes here need to be made apparent to a general readers.

We provided a broader theoretical perspective in response to R1's recommendations, please see our changes below.

Line 52-68: In this paper, we aim to explore the developmental and phylogenetic origins of the ability to use evidence to detect a hidden cause. From an evolutionary perspective, some authors have suggested that humans are the only species capable of representing and reasoning about causality (5), while others contend that we may share some mechanisms for dealing with natural causal structures, in order to infer the location of food or when using tools (6). Nevertheless, there is general agreement that some components of human causal reasoning eventually represent a significant departure from anything present in other animals, and the tendency to seek explanations for unseen events is one candidate, because it might require a kind of thinking about past possibilities scaffolded through cultural and linguistic input over human development (14,15). From a developmental perspective, the timing of emergence of this ability is therefore important, as it helps us to determine whether humans have a

natural tendency to look for and infer causal relations in their environments early in life, perhaps even from birth (16,17). If so, its development may follow a single trajectory into adulthood and the difference between younger and older children and adults may be explained by other domain general components (i.e., information processing and attentional capacities) and expanding causal knowledge. Alternatively, early reasoning abilities may undergo significant developmental change in childhood due to the cultural input from others with development of language (1,13,18).

Referee: 1

Comments to the Author(s)

The manuscript reports three experiments – one on preschoolers and two on capuchin monkeys, examining their capacity to make a causal inference from visual mechanistic information that unfolds over time. Overall, I have positive inclinations towards this manuscript, but I do have some concerns that I would like to see the authors address.

Thank you for all your very constructive comments.

1) I am not sure I understand the procedure, as it is never clearly explained. Here's what I think happens: Participants see two cups covered in foil. One cup's foil is ripped, and that cup is baited with a reward. The cups are then hidden behind a screen, and a pen is used to rip the other cups foil, so now both cups have ripped foil. Participants are then asked to retrieve the interesting object. Four-year-olds can do so; three-year-olds are at chance (Exp 1). Capuchins can also do so (Exp 2), but they cannot do this when the relation between the movement of the implement that was used to rip the foil and the causal change (the zigzags on the cup in exp 3) is arbitrary. My concern is that even though I have read this manuscript many times, I'm not 100% certain that this is the procedure, as the description of the procedure keeps changing, and is often incomplete (e.g., in the abstract, it's not clear that the baiting and the ripping aren't done (or can't be done) on the same cup. A concrete recommendation is that the authors slow down and write the procedure so clearly that there is no possible way a reader can be confused. A figure detailing the procedure would also help (Figure 1 is insufficient).

Sorry that this was not clear enough! In fact, both cups are intact at the start. There are 2 partly-hidden events, which occur in counterbalanced order. One in which the reward is hidden, and one in which a stick/pen is seen to poke downwards. After each event, the participant is shown the outcome – first 1 cover is ripped, and then after the second event both covers are ripped. So the 2 events must have occurred on different cups.

We attended to the description throughout the manuscript to make it clearer (In the Abstract, in the Intro Lines 126-134, and minor changes in the Methods). We also replaced Figure 1 with another figure that explains the procedure visually.

This concern is particularly important because I do wonder if there are associative cues that could explain these behaviors. The arbitrary experiment (Exp 3) suggests this is not the case, but it's also possible that because the capuchins need to forage to survive, the use of this kind of temporal cue has been ramped up, much like the Garcia effect in rats. Put another way, the authors conclude that this causal reasoning capacity is shared between human beings and non-human primates, but it is also possible that what the monkeys have evolved is a domain-specific associative strategy that allows them to solve this particular task (one that involves foraging), but not causal inferences from temporal

information more generally? This strikes me as a clear alternative to the authors' interpretation, which needs to be addressed.

We wouldn't want to argue from these data that monkeys have a generalized and abstract grasp of the notion that causes must precede their effects that they could use to reason in any context – especially if the timescale or causal mechanism was far removed from their natural ecology and their experience. Importantly, we would argue that neither do humans, unless they receive explicit teaching that allows them to extrapolate their experience into a symbolic/linguistic rule. We try to make this point clear in a new paragraph in the introduction (Line 52-68, copy/pasted in our response to the associate editor). The details of the representation and algorithm that make it possible for the monkeys to use the information in the ripped foil task better than they did in the arbitrary control still need to be worked out. It might be that a model built out of the associative tradition can do the job – it might be that one needs something like an 'intuitive physics' to ground this learning in. We don't think our data speak to this point and we try to make the scope of our argument clearer in the revision.

2) Assuming that I have at least some aspect of the procedure correct, the task reminds me a great deal of work by Teresa McCormack and Christoph Hoerl on causal reasoning through temporal sequencing (e.g., McCormack & Hoerl, 2005). It seems to me that the procedure used there is very similar to the procedure used here. They find that it isn't until age 5 that children succeed (they did not test non-human primates to my knowledge). It would be worthwhile for the authors to integrate the present findings with these. Related to the point that I made in #1, would the monkeys perform similarly to the older children or the younger children in this task?

We agree that this study is highly similar to the procedures used by McCormack and Hoerl experiments (2005, 2007). However, we believe our task is analogous to their "visible conditions" (Study 2 in McCormack & Hoerl, 2005 and Experiment 4 in McCormack & Hoerl, 2007) where children at least by 3-years of age succeed. In both experiments, children could observe the two events sequentially and update their model of the world as the events unfolded (e.g., "Doll 1 pressed button 1 so there must be a toy car in the window of the apparatus. Then Doll 2 pressed button 2 so the toy car must now be replaced by a marble"). Hoerl and McCormack (2019) argue that alongside young preschoolers many nonhuman animal species can also succeed in such *temporal-updating* tasks.

Similar to these "visual" conditions, in our task too, the participants can observe the experimenter picking up the objects from the table one at a time and ripping the cups sequentially. Differently from their design, in our case the visual evidence is indirect. When the order of events is reversed in the Transfer phase, they can still engage in temporal updating to locate the reward in the correct cup (if they can grasp the causal link between the poking objects and the ripping events). As opposed to the *temporal-reasoning* tasks, where only by 5-years of age children succeed (McCormack & Hoerl, 2005; 2007; Povinelli, Landry, Theall, Clark, Castille, 1999) our task does not require the participants to reason about the causal significance of the temporal order of events.

We now added a reference to this literature in our intro where we describe our design.

Line 116-125: The current task required participants to locate a reward in one of two cups covered with foil, based on the appearance of rips in the foil covers following two sequential, partially hidden baiting events. Previous research shows that children at least by 3-years of age can update their knowledge (i.e., about the location of the reward) as they receive direct visual information about events in a certain order (49,50). In our task they will be required to do so using indirect visual information. As for the capuchin monkeys, there is also evidence for the capacity to represent hidden objects: they expected to

find the same number (or type) of rewards in an apparatus as they had previously seen being hidden (51).

3) How was the sample size for the child sample determined? An a priori power analysis was not conducted, and no stopping rule was in place; in general, effect sizes are not reported, except in the figures. But with the figures small, this information is hard to decipher. The figures are also in black and white, so it is hard to tell the color coding (e.g., they refer to a red line, but I have no idea which line they are referring to, nor do I know what the vertical arrows represent).

We did not preregister our study, but we decided on our sample size a priori and stated this in our ethics application – 32 children in each age group. We did not reach our desired sample size in all age groups due to younger children not passing the pre-test phase and time constraints. More information on these is now provided in the manuscript (also please see below).

We also corrected our sample size in the manuscript. We had reported the sample size including only those who participated in the Test and the Transfer phases (N=68) and excluding those who could not pass the initial Pre-test phase (N=15). In retrospect, we believe excluding those children who did not pass the Pre-test phase does not represent our true sample size. Now you can access the data for the Pre-test phase in our GitHub repository. Table S1 in the ESM is also updated in line with this change.

Lines 151-157: Eighty-three 3-5-year-olds ($M_{age}=52.10$, $SD_{age}=9.58$) participated. Six additional children were tested but did not complete the task. Age and sex were counterbalanced as much as possible (Table S1). Our target sample size were 32 children in each age group (stated in the ethics application). Due to the opportunistic sampling approach in nurseries, time constraints, high drop-out rates in the warm-up phase in 3-year-olds (N=11), and the lower number of 5-year-olds in nurseries we only reached our target sample size with 4-year-olds. No analysis was conducted prior to the end of data collection.

It is unfortunate that there were no colors in our figures in the manuscript. To clarify, we changed the “red dashed line” to “dotted line” to indicate the chance level at 0.5 on y-axis. The arrows were used to indicate the age (in months) when the model prediction is significantly above the chance level. We wanted to visually highlight the age difference with regards to when children passed the Test and the Transfer phases. We agree that this is not common practice in figures and perhaps complicated the figure, so we removed the arrows from all of our figures in the ms and the ESM.

As for the effect sizes of the predictor variables: apart from the model predictions and confidence intervals shown in the figures we also report the estimates, standard errors and confidence intervals of the predictors in the electronic supplementary material (Table S2-S12). Unfortunately, we cannot add them to main manuscript due to the page limit.

4) In the developmental literature, there are several possible ways to interpret the age-related change. The authors discuss some of them (e.g., cognitive control and demand characteristics of the procedure). I agree that demand characteristics are not a strong consideration, and that cognitive control might affect performance, but in many of their citations, there is a third alternative, which is that between the ages of 3-4, children develop an explicit concept that causal relations have mechanisms. That is, they develop an explicit domain-general capacity that posits how events relate to one another. This could be further discussed, particularly in light of whether such a capacity is shared by non-human primates.

We added a new paragraph where we discuss this in more detail.

Lines 527-539: Another possibility is the emergence of a more abstract notion of causal mechanism. Children between the ages 3- and 4-years are known to have a developing knowledge about how events are related to their outcomes (16,19,20). It is likely that their understanding is limited to certain mechanisms they have direct experience with (such as toys working with batteries, 23) and over time they begin to generalize this knowledge and develop an explicit understanding that all causes lead to their effects through a mechanism. One route to doing this might be through the use of language. Especially in cases where children do not have access to the causal information visually, verbal input from others (e.g., instructions, explicit teaching, alerting to the presence of a mechanism, requests for explanations) becomes a valuable source of information (11,12). Once children learn that the causes of certain events may not be directly observable, they may form a new concept, 'hidden causes', which may act as a placeholder for further learning (13). This would not necessarily specify the cause of a particular event, but it could motivate children to look for causes when faced with indirect evidence.

To conclude, this is an interesting manuscript. I would like to see more clarity in the methods and argumentation to be certain that there are not alternate explanations for the phenomenon. Given the difficulty I had in figuring out the exact nature of the procedure, I would like to also see a better integration with some of the existing research that I mentioned in this review.

Thank you very much for your comments.

Referee: 2

Comments to the Author(s)

This is a thorough, interesting, and timely paper on the use of visual information by human children and Capuchins to find rewards after hidden interactions between objects. The clever design rules out various confounds by having the final state of the hiding locations look identical when revealed to participants, meaning they must use information previously acquired to pass. Similarly, the "transfer" concept is an effective way of ruling out simple arbitrary rule accounts of the behavioral patterns. I would also like to commend the authors for their efforts to keep the task minimalist with similarly low verbal instruction load between the children and the capuchins, and for making their data and code available online – which I found helpful in writing this review. I do have a number of questions and some concerns that I believe need to be addressed before this paper can be published, which I detail below:

Thank you very much for your kind comments about our paper and design.

Introduction. A point of clarification: is it correct to say about Capuchin feeding activity that "These are costly activities in terms of energy, and animals should engage in them only when it is highly likely that they will get a return"? The idea that the Capuchins should ONLY engage in seeking out food when returns are highly likely seems counterintuitive – is there a citation for this, or perhaps does the claim need to be tempered somewhat?

We agree this sounds counterintuitive as they always need to forage. We replaced this sentence and stressed the importance of making use of indirect cues in unstable environments.

Lines 104-106: In such unstable environments, the ability to make inferences about hidden causes based on incomplete information would be of potential benefit (e.g., tapping on the nut to infer its fullness or weighing a tool before using it to crack a nut) (44).

Methods.

Could the authors please clarify the following points. Was the study pre-registered? How was sample size determined in each study? What was the stopping rule for data collection? Was a power analysis conducted? More information about these sample considerations would be helpful in the manuscript.

We did not preregister our study, but we decided on our sample size a priori (we did not conduct a power analysis) and stated this in our ethics application – 32 children in each age group. We did not reach our desired sample size in all age groups due to younger children not passing the pre-test phase and time constraints. More information on these is now provided in the ms.

We also corrected our sample size in the ms. We had reported the sample size including only those who participated in the Test and the Transfer phases (N=68), and excluding those who could not pass the initial Pre-test phase (N=15). In retrospect, we believe excluding those children who did not pass the Pre-test phase does not represent our true sample size. Now you can also access this data in our GitHub repository. Table S1 in the ESM is also updated in line with this change.

Lines 151-157: Eighty-three 3-5-year-olds ($M_{age}=52.10$, $SD_{age}=9.58$) participated. Six additional children were tested but did not complete the task. Age and sex were counterbalanced as much as possible (Table S1). Our target sample size were 32 children in each age group (stated in the ethics application). Due to the opportunistic sampling approach in nurseries, time constraints, high drop-out rates in the warm-up phase in 3-year-olds (N=11), and the lower number of 5-year-olds in nurseries we only reached our target sample size with 4-year-olds. No analysis was conducted prior to the end of data collection.

The various criteria for passing or failing, moving on from pre-test to test phases, and so-on felt somewhat arbitrary. Could the authors please specify whether these were all determined a-priori, or if they were changed at all after the data had been collected?

Initially, we had planned to run 10 trials of the pre-test phase and the criterion was thought to be 5 consecutive correct trials in line with our previous study using auditory cues instead of visual cues (Civelek et al., 2020). However, in our pilot, since children did very well in this phase and got bored, we decided to reduce the number of trials: 5 in total, with 2 consecutive trials correct as our learning criterion. We also initially planned to run 10 trials per Test and Transfer phases but reduced this to 8 trials to shorten the length of the session. These decisions were all done a-priori and analyses were only run after data collection was completed.

For example, why exactly was the pre-test phase so different for the Capuchins and the children? And why exactly did the authors make the following change between E1 and E2 (children vs monkeys): “In the main analyses, we used the last 16 trials of Test phase for each monkey (as opposed to all trials) and the 16 trials (2 sessions) of the Transfer phase. Since we were interested in the potential performance change from Test to Transfer, we wanted to include the monkeys’ best performance in the Test phase.”

Wouldn't the same logic apply in both cases?

We faced different limitations when testing capuchin monkeys and children. For capuchin monkeys we could test them over multiple test sessions, but our sample size was confined by the number of monkeys at the research site. For the children, there was no strict sample size limit but we could not test them over multiple sessions. The changes between Exp 1 and Exp 2 reflect these limitations.

We added the following paragraph to our procedure with the monkeys to highlight the differences in terms of procedure between children and monkeys:

Lines 307-316: The procedure for the monkeys differed from children in the following ways: 1) The monkeys received an additional Warm-up for familiarization. The monkeys had extensive experience in finding a reward hidden in cups (55)– so this phase was just to ensure that the novel materials (tall colourful cups) did not cause any neophobia. The monkeys were able to pass this phase with no difficulty. 2) Monkeys received 8 trials per session and the criterion to pass a phase was 14/16 trials correct (binomial test: $p=0.001$) in two consecutive sessions. 3) We decided (also due to our limited sample size) that if monkeys did not perform at above chance levels in the first 16 trials, they would receive further trials (Table S7) to see if they could learn the task before moving on to the last phase where we evaluated their ability to flexibly transfer to a reversed order of events.

When it comes to fitting the models, the authors specify that they “left out the correlation parameters between random intercepts and random slopes terms” – was this because the lmer model did not converge without this specification? Could the authors please clarify in text their reason for this modeling decision.

We decided a priori not to include the random correlations to avoid an over-specified random effect structure (Bates et al. 2018) and also given that simulations studies found that models without random correlations basically have the same type 1 error rate than models including random correlations (Barr et al., 2013). We added this information to the electronic supplementary material (Lines 30-33).

Similarly, in the GLM 01 in Experiment 1 and elsewhere (experiment 3) the notes in the R script says there is a singular fit warning for some models. If there was a singular fit for given models, did the authors attempt to rectify this issue and if so, could they please explain how? If they believe the singular fits for the lmer mixed effects models are not a problem, could they please clarify.

In all cases in which we encountered singular fit messages, we simplified the random slope structure until the singular fit message disappeared or all random slopes were removed. You can now find these analyses in ripchild.Rmd (Lines 514-523) and ripmon.Rmd (Lines 516-530), common.Rmd (Lines 492-502). In no case did this affect the pattern of significant results. We decided to stick to the model with all random slopes included because it is well established that neglecting random slopes can lead to an elevated type I error rate (Barr et al. 2013). We clarified this point in the supplementary material (Line 37-40).

Discussion

I do not follow the authors explanation for the fact that the monkeys performed better in the food-stick condition. The authors write: “differing memory demands associated with these trial types seem to be a plausible explanation” and then mention executive functions – but unless I am missing something they do not elaborate, and I cannot figure out why this would be a plausible explanation for the difference.

We have now elaborated on this in the following lines:

Lines 413-422: In the food-stick trials, once the subject saw one cup was ripped after the food reward was picked up, it did not need to pay attention to the rest of the trial to solve the task, provided it could remember the location of the first ripped foil. Indeed, we observed that in food-stick trials, after the first cup was revealed to be ripped (by the food), the monkeys would often move on to that side of the

cubicle and extend their arm through the hole in the window until the end of the trial without paying any attention to the experimenter's next actions (i.e., picking up the stick to rip the other cup). On the other hand, in the stick-food trials, they had to remember which cup was ripped initially to locate the reward in the correct cup after the food hiding event. Paying attention only to the result of the food hiding event would not lead to success because by then, both foils are ripped.

The conclusion states: "without the need for verbal instruction in children and nonhuman primates" but this may need to be slightly re-worded as there was of course some verbal instruction for the children.

Thank you. We changed the wording here to make it clearer.
Line 521: verbal prompts to look for and identify causes

Referee: 3

Comments to the Author(s)

This paper reports an interesting investigation of human children's and capuchin monkeys' physical reasoning abilities to infer causes from visual effects that extends previous research by using a novel paradigm. The study design allows to differentiate between participants understanding of physical causes or participants using arbitrary association rules to choose between reward locations. The authors find that 4-year olds but not 3-year olds solved the experimental task successfully and they performed above chance already in their first trials. With increasing age, children also became better at reporting correct explanations for the events during the experimental procedure. On the group level, capuchin monkeys performed above chance when considering all trials. They had more difficulty to solve the task with arbitrary cause-effect relations (Exp. 3) than with causally meaningful cause-effect relations (Exp. 2).

Overall, the submission is in very good shape and was a pleasure to read. I found the manuscript easy to follow, the study is well embedded in existing relevant literature, supplementary material is appropriately referenced in the main body, and data & analysis code are available. Experimental procedure and statistical approach are in principle suited to test the hypotheses, but I have some comments regarding the relationship of individual pass/fail performance and assessment of group-level performance in the monkey experiments (see below). This issue requires clarification (either when introducing the success criterion or in the discussion or both) or a more consistent data assessment approach. I only have a few other comments, which I am confident the authors will be able to address.

Thank you very much for your kind words and helpful comments.

ESM

- ESM 34-37 Why was the effect of age assessed in a separate model comparison?

This section (in the current 'track changes' doc lines 45-48) refers to whether age had an effect on children's ability to explain how they found the sticker at the end of the task. We added an 'open-ended question' description to highlight it.

- ESM Table S2: Why do you provide χ^2 and df values for some but not other terms (applies to this and other tables)? I found the combination of table notation and footnote a bit hard to read and recommend to only name the term (e.g. "Trial type") bar the reported category ("sticker-pen") in the

table rows. Especially the rows with interaction terms would become easier to read. By providing information on reference categories in the note (as you already do) all information is available.

We revised all tables in the ESM and clarified what is reported. We had mixed p values from wald tests (intercept, main effects of predictors that are part of an interaction) and p-values from likelihood ratio tests (LRTs). We only based our inferences based on LRTs which are seen as more reliable. Therefore, we report in the revised ESM only the LRT p-values.

The R function `drop1` that we used to calculate LRTs for the fixed effects removes one term at a time from the model and compares the reduced model to the full model. In the presence of an interaction term, `drop1` only removes the interaction term from the model but not the predictor variables it encompasses (taking out the main effect in the presence of an interaction term would not be meaningful). If the interaction term was significant, one should not interpret the main effect anyway. If the interaction term was *not* significant, we would remove the interaction term to be able to evaluate the main effects.

We agree that the rows were a bit crowded and we removed the reference categories from the tables. They remain in the footnotes.

Experiment 2

- L 250-255: Where there any monkeys who didn't pass the pre-test?

No, they all passed the pretest. This is now written more clearly in the manuscript.

Lines 325-326: All the monkeys passed this phase within 2 to 6 sessions.

- L 243-259 The "pass" criterion is not fully clear to me both in terms of its application and in terms of its pass/fail consequences. Could "14 of 16 trials correct" be reached across three sessions or did the monkeys have to perform at this level in two consecutive sessions? It also appears that no monkey was ever excluded from proceeding to the next stage when failing to reach the criterion?

The criterion was to reach 14 out of 16 trials correct in two consecutive sessions and this is now clarified in the manuscript. And correct, we did not exclude the monkeys for not having reached the criterion, they always moved on to the Transfer phase.

Lines 311-312: Monkeys received 8 trials per session and the criterion to pass a phase was 14/16 trials correct (binomial test: $p=0.001$) in two consecutive sessions. And later in Lines 324-325: Monkeys could receive up to 10 sessions in total if they did not reach the criterion sooner.

- It came as a surprise to me that on the one hand, 15 of 19 individuals didn't reach the criterion in the Test condition at all, but on the other hand group performance was above chance. Doesn't this mean the set criterion is not a good indicator of an individual's task performance? I find it a bit odd that on the one hand, individuals are categorized as having "passed" a pre-defined criterion and are not further tested in this test phase (they move on to the transfer phase immediately), while on the other hand other individuals have not passed the criterion but are not categorized as "failed"; instead their data of 80 trials is used to analyse group performance whereas we don't know how the "successful" individuals would have performed in the remaining sessions. Speaking from own experience, I repeatedly encountered that passing a numerical criterion doesn't mean that a monkey keeps performing at above chance levels if tested further. The authors' approach thus seems a bit inconsistent here and requires

more explanation and discussion of function and consequences of the pre-defined criterion and how it relates to the reported group performance.

Our criterion was based on a binomial test (while a binomial test would already be significant with 13/16 correct, we wanted to be more conservative to account for multiple testing issues). In any event, the group-level analysis based on an intercept-only model gave us more power, which explains the apparently discrepant findings between the individual-level and group-level analyses.

The GLMM that we used for analyzing the data is based on trial-level performance. It includes the random effect of subject ID and random slopes of trial number. Therefore, the model takes into account that some individuals only have data for the first 16 trials. We deem it unlikely that these four individuals would have pulled the group performance down to the chance level had we administered all 80 trials with them (which would have only happened if they had performed *below* 0.5 in the remaining sessions).

However, we would also like to emphasize that we implemented the criterion mainly for a different reason, namely to avoid overtraining the monkeys. We were mainly interested in their transfer phase performance. For this purpose, we wanted to give them enough experience with one order of events but without overtraining them.

- Also, I 302-308 should be moved further up (around I 279-282) so the reader immediately learns that most individuals failed to reach the criterion despite above chance group level performance.

Thank you, we moved this to the top of the results section now.

Lines 355-359: We analyzed the monkeys' performance in the first 8 trials of the Test phase. It did not differ from chance (0.16 ± 0.17 , $z = 0.93$, $p = 0.35$). Only 1 out of 19 monkeys performed above chance level in the first 8 trials of the Test phase according to a two-tailed binomial test ($p = .001$). We therefore proceeded to give the monkeys further trials to explore their ability to learn to solve this task, and then transfer their knowledge.

- It would be interesting to learn if performance of "successful" individuals differs from that of "failed" individuals in the following transfer phase.

We ran this analysis now (Lines 535-571 in ripmon.Rmd file in github). There is no difference between the Transfer performance of those who reached the criterion in the Test phase and those who did not.

Lines 400-402: We further found that the Transfer performance of the monkeys that reached the criterion in the Test phase ($N=4$) did not differ from those that did not reach the criterion in the Test phase ($N=15$) ($\chi^2(2)=5.45$, $p=.065$, Table S10).

- To complement performance assessment of all trials for the whole group, the authors might also want to report how many of the 15 individuals who never reached the numerical 14 of 16 criterion performed above chance when all their trials were considered.

We ran binomial tests for each of the 15 individuals when considering all 80 trials per individual in the test phase. 5 out of 15 individuals performed significantly above chance level. (Lines 582-716 in ripmon.Rmd file in github).

Line 359-364: The number of sessions needed to pass the Test phase ranged from a minimum of 2 (16 trials) to a maximum of 10 (80 trials). Most of the monkeys (15 of 19) completed the maximum number

of sessions before they moved on to the Transfer phase without meeting criterion (Table S7). The binomial tests for each of the 15 'unsuccessful' individuals showed that when considering all 80 trials in the Test phase, 5 of them performed significantly above chance.

- L 307-308: How can a monkey reach a criterion of 14 of 16 correct choices (as described in I 256/257) after only 8 trials?

We rephrased this sentence to correct our mistake.

Lines 356-358: Only 1 out of 19 monkeys performed above chance level in the first 8 trials of the Test phase according to a two-tailed binomial test ($p = .001$).

Experiment 3

- L 333: Typo -> delete "if they"

Corrected.

Appendix B

Associate Editor

Note from the authors: The line numbers in this response refer to the tracked changed (i.e., revised) document.

The reviewers and I agree that the authors have done a nice job addressing the points raised in the previous round of review, which has clarified many key aspects of the paper and strengthened the rationale of the work.

Thank you very much for your positive feedback!

The reviewers have a few final comments clarifying aspects of the paper and its interpretation. One thing to note is the comment from R1 concerning a diagram for study 3. Along those lines, I will also note that while the current diagram is quite helpful, portions of it are extremely small and what I believe is intended to be an image of a sticker appears to be a tiny figure (only by zooming in did I see it was an animal, which is somewhat confusing). A second is the important point from R2 concerning the difference between understanding the outcome one's own manipulations on the world, versus learning about causality via observation. Given the focus in comparative work on the role of tool use in spurring the emergence of such cognitive skills, it would be worth it to mention this distinction.

I would also suggest a bit more nuance in the discussion of the capuchin results. The general discussion states that "capuchin monkeys did not show convincing evidence for this ability, as they performed at chance in the first trial of transfer task." While is true, capuchins also performed quite well overall in the transfer; their trajectory in the transfer seems different than in the original test (e.g., they do not slowly re-learn this relationship; their performance jumps on trial 2); and study 3 supports the claim that they are not just using associative learning mechanisms. In that sense, they seem to show some differences from the younger children as well.

- 1) We now added another figure (Line 420-424) to visualize the procedure for Experiment 3 and updated our Figure for Experiment 1&2 (Line 162-170) with the help of our Design team. We made the foil and paper covers to look like lids, used a star to represent the reward (added this information in the text to make it clearer) and a diagonal arrow to represent the temporal progression of events in each phase. We made our opaque barrier appear semi-transparent in the figure for the reader (so that they can see how the events took place behind the barrier, as well as the reward and the stick).
- 2) We now changed the examples to fit our task better and, in this sentence, we also highlighted the difference between learning via naturalistic observations vs one's own manipulations as in operant conditioning.
Line 94-99: In such unstable environments, the ability to make inferences about hidden causes based on incomplete observational evidence (e.g., inferring that a nut is empty based on wormholes on its shell) would allow the monkey to go beyond learning only about causal relations that they can manipulate (e.g., tapping on the nut to infer its fullness or weighing a tool before using it to crack a nut) (39).
- 3) Thank you for this comment. We refer to the difference between Experiment 2 and 3 towards the end of the General Discussion but we agree that this is an important finding that should be highlighted at the beginning of the discussion too. We replaced the 'convincing evidence' with 'robust evidence' and added the following part:

Line 455-459: However, given additional experience capuchin monkeys were able to transfer their knowledge to the reversed order of events only when the visual traces were causally related to the food's location (Experiment 2) as opposed to when they were arbitrarily related (Experiment 3). This suggests that causal representations may play a role in their problem solving.

However, we would refrain from suggesting that capuchins performed better than 3-year-olds as the number of trials they received are not comparable. It is highly likely that with the same number of trials as capuchins, 3-year-olds would have performed similarly in the Transfer phase.

Finally, we made some minor changes throughout the ms to stay within the word limit.

Thank you again for all the suggestions.

Referee: 1

Comments to the Author(s).

This is a revised version of a manuscript I reviewed previously (I was Reviewer X). My main concern previously was with the clarity of the manuscript and experiment, and the authors have done an excellent job improving the manuscript with this goal in mind. I do have some minor remaining concerns, but in general, I am highly satisfied with this revision.

Thank you very much.

1) I did not understand what was being analyzed in lines 240-242 under the phrase "open ended analysis" – a sentence or two to roadmap the reader might help here.

At the end of the task, we asked children how they found the sticker. We were interested to see whether children would refer to their 'strategy' explicitly when asked an open-ended question (e.g., referring to the ripping with a sticker and a pen, saying they chose the blue cup because they liked its color). Therefore, we named this variable "open-ended question" but notice that this may not be a good reminder for the reader in the results section, so we changed its name to "explanation question" throughout the ms and the ESM. We also added a reminder for what this question is.

Line 251-254: Explanation question. Only 16 children gave the correct explanation to the experimenter's question "How did you decide which cup to choose?" at the end of the task. Older children gave the correct explanation significantly more often than younger children (GLM: $\chi^2(1)=5.15$, $p<.05$) (Table S5). None of the children below 48 months of age gave the correct explanation (Figure 2).

2a) Much in the same way that the addition of the text in Figure 1 (And the better clarity of the figure itself) facilitated an understanding of the procedure in Experiments 1-2, a figure for Experiment 3 would allow the reader to understand the distinction between Experiment 2 and 3 better. To be perfectly honest, I did not understand what was specifically different about Experiments 2 and 3, and the manuscript suffers from a lack of clarity here.

We now updated our figures with the help of our Design team to make the procedure clearer for both Experiments 1, 2, and Experiment 3. Specifically, we made the foil and paper covers to look like lids, used a star to represent the reward (added this information in the text to make it clearer) and a diagonal arrow to represent the temporal progression of events in each phase. We made our opaque barrier

appear semi-transparent in the figure for the reader (so that they can see how the events took place behind the barrier, as well as the reward and the stick).

The only difference between the two experiments is that instead of using foil covers that can be ripped, we used white paper lids that are not ripped but replaced with black and white zigzagged paper lids after the hiding and the poking events. This allowed us to keep the 'state change after picking up the food' element intact in both experiments (which can be used as an associative cue to solve the task) while removing the critical physical causality. We tried to stress this difference in between the experiments in the intro to Experiment 3.

Line 386-393: Instead of covering the cups with foil as in Experiment 1&2, in this experiment E covered them with white paper, and instead of the state change following the event behind the barrier being that one of the foil covers was ripped, it was that one by one these white covers were exchanged for covers painted with black zigzags. If the monkeys represented the causal relationship between the state change in foil and the food's hiding, we expected weaker performance when the cues were arbitrarily related to the location of the reward.

2b) I also wonder if a comparison between Experiments 2-3 is warranted, given that there seems to be some suggestion that performance on these measures should be different.

We would refrain from a statistical comparison between Experiment 2 and 3 as we were mainly interested in the qualitatively different pattern of results within the two experiments (i.e., evidence of transfer in one and not the other)- which we tested for. A quantitative comparison would also be problematic given that we conducted the experiments in a fixed order. It would be impossible to tease apart the differences between the experiments and order effects.

Finally – and I will admit that this is more a musing than a direct suggestion – I wonder to what extent children have statistical learning capacities early on, whilst their causal reasoning capacities emerge from those capacities and are limited by information processing demands, which might motivate the difference between the younger and older children in the sample. That is, if those information processing demands are reduced, or if children have more explicit knowledge to allow them to engage in retrospective reasoning, then 3-year-olds would look more like the older children in the sample (but the non-human primates would not show such a benefit, indicating a uniquely human system – although if they do, then that would speak against such an argument).

Again, this is musing, but to give concrete feedback, I would have liked to see more of a discussion of the possible mechanisms of development in the GD, particularly looking at places where children show the developmental difference between 3-4. What do those researchers suggest is developing and does that relate here? My guess is that this is more about the set-up of the experiment and the inference that children have to make about changes of state than about retrospective inferences, per se. Again, I feel like Experiment 3 can shed some insight on this idea, but because it was not as clearly presented as the other two experiments, it was more difficult to tell.

As Woodward (2011) claims, we believe that causal reasoning is not a unitary capacity but rather emerges from an interaction of a set of abilities that are well integrated in human adults. Some of these skills such as statistical learning, and some domain specific sensitivity to certain causal relations are present even in infancy; and the general processing capacity develops significantly during preschool

years. We also believe that the cultural input (e.g., formal education, explicit instructions, requests to explain) that comes with the development of language has a significant influence on children's causal reasoning development. All of these may explain the commonly found difference between the 3- and 4-year-olds' performance and future work could shed light on this even more. We did not test for the role of causal language, nor of executive functions in this study so we did not want to speculate more than we did- instead we discussed all possibilities. We now added a sentence in GD to clarify this.

Line 466-469: As causal reasoning is not a unitary ability but rather an emergent capacity resulting from the interplay of distinct abilities (e.g., statistical learning, domain-general processing capacity, domain-specific knowledge of causal relations, cultural input) that develop significantly during preschool years, several possibilities could explain this difference.

To be more specific, the literature suggests that while children have statistical learning capacities early on, even by 8-months of age (Aslin et al., 1998; Saffran et al., 1996), this capacity does not consistently support causal predictions from natural covariations until around 4-years-of age in the absence of additional causal cues (e.g., the presence of a goal-directed agent, causal instructions or questions used by the experimenter). We addressed this issue in previous research -with reference to the role of verbal scaffolding- in the introduction (lines 68-77). And it is also true that 3-year-olds perform better in causal reasoning tasks when they already have explicit knowledge about the causal relations in question (e.g., Buchanan & Sobel, 2011: wires vs. batteries; Bullock, Gelman, & Baillargeon, 1982: rolling balls vs. light). However, successful performance in tasks with familiar objects/relations makes it problematic in terms of teasing apart the role of previous experience and reasoning as children might be relying on an arbitrary rule such as "Batteries = active toy". We used a familiar causal relation in our task as well – ripping by poking – but the critical causal event was hidden in our study so we could assess whether children relied on their prior knowledge and inferred/imagined a causal relationship from the effects. We did not give young children any experience with ripping materials (e.g., tissues) beforehand as we were primarily interested in their spontaneous performance. However, if children lacked this knowledge, it may have been difficult for them to succeed in the task within 8 trials. We now acknowledge the role of this type of explicit knowledge on causal reasoning abilities of young children in the GD.

Line 476-479: However, if young children lacked this knowledge, it would have been challenging to pass the task within the 8 trials they were given. To explore this further, 3-year-olds could be provided with additional experience (e.g., poking tissues or paper with sticks) to see if they generalized this knowledge to the experimental context.

We also addressed the development of executive functions as a possible explanation for 3-year-olds' performance in this task. We think the processing demands of our tasks was very low -only keeping track of two events where information can be updated each time the barrier is lifted (e.g., first ripped with sticker, then with pen). We know from over-imitation studies that children of this age are very well able to imitate a sequence of events that shows the ability to track and remember event sequences. However, we did not test this explicitly. We added a sentence to make this clearer.

Line 485-486: In this study, we did not assess children's ability to keep track of two events in the absence of inference-making requirement (but see, 59).

To conclude, again, this is a strong revision and a very nice piece of work.

Thank you very much!

Referee: 2

Comments to the Author(s).

I believe the authors have sufficiently addressed my initial round of comments, as well as the various important issues raised by the other reviewers. I have only one minor follow-up query about the new examples in the introduction, which seem somewhat detached from the phenomenon that the experimental protocol assesses. The authors write: “(e.g., tapping on the nut to infer its fullness or weighing a tool before using it to crack a nut)” – though both of these examples involve the animal making some kind of environmental manipulation in order to gather information. In the experiments here, the animal witnesses a hidden event and is then presented with the opportunity to act upon an inferred causal relationship that took place during that hidden event. Are there naturalistic examples that more closely parallel this structure than the “tapping on the nut” or “weighing a tool” examples? Perhaps I am misunderstanding either the examples or the task itself, and if so, I would appreciate any clarification that the authors could provide.

Thank you very much for pointing this out. It is true that what we are aiming to assess in this paper is whether the subjects can make inferences based on ‘naturalistic’ observations of hidden events rather than learning based on ‘egocentric’ feedback/acting on the objects themselves. That is also why we did not provide our participants with prior experience with poking and ripping objects in advance. In the introduction, we chose these examples with the existing research in mind about capuchin monkeys’ capabilities with regards to hidden causes; however, we agree that these examples do not represent what we are measuring in our task, so we changed them accordingly. We also stressed the difference between learning from naturalistic observations vs. one’s own manipulations in line with AE’s suggestion.

Line 94-99: In such unstable environments, the ability to make inferences about hidden causes based on incomplete observational evidence (e.g., inferring that a nut is empty based on wormholes on its shell) would allow the monkey to go beyond learning only about causal relations that they can manipulate (e.g., tapping on the nut to infer its fullness or weighing a tool before using it to crack a nut) (40).

I think this paper will make a useful and timely addition to the literature on the developmental and evolutionary roots of causal understanding. Thank you for the opportunity to review this interesting research and best wishes moving forward with this line of work.

Thank you very much for the kind words and encouragement!

Appendix C

Associate Editor

Note from the authors: The line numbers in this response refer to the tracked changed (i.e., revised) document.

The reviewers and I agree that the authors have done a nice job addressing the points raised in the previous round of review, which has clarified many key aspects of the paper and strengthened the rationale of the work.

Thank you very much for your positive feedback!

The reviewers have a few final comments clarifying aspects of the paper and its interpretation. One thing to note is the comment from R1 concerning a diagram for study 3. Along those lines, I will also note that while the current diagram is quite helpful, portions of it are extremely small and what I believe is intended to be an image of a sticker appears to be a tiny figure (only by zooming in did I see it was an animal, which is somewhat confusing). A second is the important point from R2 concerning the difference between understanding the outcome one's own manipulations on the world, versus learning about causality via observation. Given the focus in comparative work on the role of tool use in spurring the emergence of such cognitive skills, it would be worth it to mention this distinction.

I would also suggest a bit more nuance in the discussion of the capuchin results. The general discussion states that "capuchin monkeys did not show convincing evidence for this ability, as they performed at chance in the first trial of transfer task." While is true, capuchins also performed quite well overall in the transfer; their trajectory in the transfer seems different than in the original test (e.g., they do not slowly re-learn this relationship; their performance jumps on trial 2); and study 3 supports the claim that they are not just using associative learning mechanisms. In that sense, they seem to show some differences from the younger children as well.

- 1) We now added another figure (Line 420-424) to visualize the procedure for Experiment 3 and updated our Figure for Experiment 1&2 (Line 162-170) with the help of our Design team. We made the foil and paper covers to look like lids, used a star to represent the reward (added this information in the text to make it clearer) and a diagonal arrow to represent the temporal progression of events in each phase. We made our opaque barrier appear semi-transparent in the figure for the reader (so that they can see how the events took place behind the barrier, as well as the reward and the stick).
- 2) We now changed the examples to fit our task better and, in this sentence, we also highlighted the difference between learning via naturalistic observations vs one's own manipulations as in operant conditioning.
Line 94-99: In such unstable environments, the ability to make inferences about hidden causes based on incomplete observational evidence (e.g., inferring that a nut is empty based on wormholes on its shell) would allow the monkey to go beyond learning only about causal relations that they can manipulate (e.g., tapping on the nut to infer its fullness or weighing a tool before using it to crack a nut) (39).
- 3) Thank you for this comment. We refer to the difference between Experiment 2 and 3 towards the end of the General Discussion but we agree that this is an important finding that should be highlighted at the beginning of the discussion too. We replaced the 'convincing evidence' with 'robust evidence' and added the following part:

Line 455-459: However, given additional experience capuchin monkeys were able to transfer their knowledge to the reversed order of events only when the visual traces were causally related to the food's location (Experiment 2) as opposed to when they were arbitrarily related (Experiment 3). This suggests that causal representations may play a role in their problem solving.

However, we would refrain from suggesting that capuchins performed better than 3-year-olds as the number of trials they received are not comparable. It is highly likely that with the same number of trials as capuchins, 3-year-olds would have performed similarly in the Transfer phase.

Finally, we made some minor changes throughout the ms to stay within the word limit.

Thank you again for all the suggestions.

Referee: 1

Comments to the Author(s).

This is a revised version of a manuscript I reviewed previously (I was Reviewer X). My main concern previously was with the clarity of the manuscript and experiment, and the authors have done an excellent job improving the manuscript with this goal in mind. I do have some minor remaining concerns, but in general, I am highly satisfied with this revision.

Thank you very much.

1) I did not understand what was being analyzed in lines 240-242 under the phrase "open ended analysis" – a sentence or two to roadmap the reader might help here.

At the end of the task, we asked children how they found the sticker. We were interested to see whether children would refer to their 'strategy' explicitly when asked an open-ended question (e.g., referring to the ripping with a sticker and a pen, saying they chose the blue cup because they liked its color). Therefore, we named this variable "open-ended question" but notice that this may not be a good reminder for the reader in the results section, so we changed its name to "explanation question" throughout the ms and the ESM. We also added a reminder for what this question is.

Line 251-254: Explanation question. Only 16 children gave the correct explanation to the experimenter's question "How did you decide which cup to choose?" at the end of the task. Older children gave the correct explanation significantly more often than younger children (GLM: $\chi^2(1)=5.15$, $p<.05$) (Table S5). None of the children below 48 months of age gave the correct explanation (Figure 2).

2a) Much in the same way that the addition of the text in Figure 1 (And the better clarity of the figure itself) facilitated an understanding of the procedure in Experiments 1-2, a figure for Experiment 3 would allow the reader to understand the distinction between Experiment 2 and 3 better. To be perfectly honest, I did not understand what was specifically different about Experiments 2 and 3, and the manuscript suffers from a lack of clarity here.

We now updated our figures with the help of our Design team to make the procedure clearer for both Experiments 1, 2, and Experiment 3. Specifically, we made the foil and paper covers to look like lids, used a star to represent the reward (added this information in the text to make it clearer) and a diagonal arrow to represent the temporal progression of events in each phase. We made our opaque barrier

appear semi-transparent in the figure for the reader (so that they can see how the events took place behind the barrier, as well as the reward and the stick).

The only difference between the two experiments is that instead of using foil covers that can be ripped, we used white paper lids that are not ripped but replaced with black and white zigzagged paper lids after the hiding and the poking events. This allowed us to keep the 'state change after picking up the food' element intact in both experiments (which can be used as an associative cue to solve the task) while removing the critical physical causality. We tried to stress this difference in between the experiments in the intro to Experiment 3.

Line 386-393: Instead of covering the cups with foil as in Experiment 1&2, in this experiment E covered them with white paper, and instead of the state change following the event behind the barrier being that one of the foil covers was ripped, it was that one by one these white covers were exchanged for covers painted with black zigzags. If the monkeys represented the causal relationship between the state change in foil and the food's hiding, we expected weaker performance when the cues were arbitrarily related to the location of the reward.

2b) I also wonder if a comparison between Experiments 2-3 is warranted, given that there seems to be some suggestion that performance on these measures should be different.

We would refrain from a statistical comparison between Experiment 2 and 3 as we were mainly interested in the qualitatively different pattern of results within the two experiments (i.e., evidence of transfer in one and not the other)- which we tested for. A quantitative comparison would also be problematic given that we conducted the experiments in a fixed order. It would be impossible to tease apart the differences between the experiments and order effects.

Finally – and I will admit that this is more a musing than a direct suggestion – I wonder to what extent children have statistical learning capacities early on, whilst their causal reasoning capacities emerge from those capacities and are limited by information processing demands, which might motivate the difference between the younger and older children in the sample. That is, if those information processing demands are reduced, or if children have more explicit knowledge to allow them to engage in retrospective reasoning, then 3-year-olds would look more like the older children in the sample (but the non-human primates would not show such a benefit, indicating a uniquely human system – although if they do, then that would speak against such an argument).

Again, this is musing, but to give concrete feedback, I would have liked to see more of a discussion of the possible mechanisms of development in the GD, particularly looking at places where children show the developmental difference between 3-4. What do those researchers suggest is developing and does that relate here? My guess is that this is more about the set-up of the experiment and the inference that children have to make about changes of state than about retrospective inferences, per se. Again, I feel like Experiment 3 can shed some insight on this idea, but because it was not as clearly presented as the other two experiments, it was more difficult to tell.

As Woodward (2011) claims, we believe that causal reasoning is not a unitary capacity but rather emerges from an interaction of a set of abilities that are well integrated in human adults. Some of these skills such as statistical learning, and some domain specific sensitivity to certain causal relations are present even in infancy; and the general processing capacity develops significantly during preschool

years. We also believe that the cultural input (e.g., formal education, explicit instructions, requests to explain) that comes with the development of language has a significant influence on children's causal reasoning development. All of these may explain the commonly found difference between the 3- and 4-year-olds' performance and future work could shed light on this even more. We did not test for the role of causal language, nor of executive functions in this study so we did not want to speculate more than we did- instead we discussed all possibilities. We now added a sentence in GD to clarify this.

Line 466-469: As causal reasoning is not a unitary ability but rather an emergent capacity resulting from the interplay of distinct abilities (e.g., statistical learning, domain-general processing capacity, domain-specific knowledge of causal relations, cultural input) that develop significantly during preschool years, several possibilities could explain this difference.

To be more specific, the literature suggests that while children have statistical learning capacities early on, even by 8-months of age (Aslin et al., 1998; Saffran et al., 1996), this capacity does not consistently support causal predictions from natural covariations until around 4-years-of age in the absence of additional causal cues (e.g., the presence of a goal-directed agent, causal instructions or questions used by the experimenter). We addressed this issue in previous research -with reference to the role of verbal scaffolding- in the introduction (lines 68-77). And it is also true that 3-year-olds perform better in causal reasoning tasks when they already have explicit knowledge about the causal relations in question (e.g., Buchanan & Sobel, 2011: wires vs. batteries; Bullock, Gelman, & Baillargeon, 1982: rolling balls vs. light). However, successful performance in tasks with familiar objects/relations makes it problematic in terms of teasing apart the role of previous experience and reasoning as children might be relying on an arbitrary rule such as "Batteries = active toy". We used a familiar causal relation in our task as well – ripping by poking – but the critical causal event was hidden in our study so we could assess whether children relied on their prior knowledge and inferred/imagined a causal relationship from the effects. We did not give young children any experience with ripping materials (e.g., tissues) beforehand as we were primarily interested in their spontaneous performance. However, if children lacked this knowledge, it may have been difficult for them to succeed in the task within 8 trials. We now acknowledge the role of this type of explicit knowledge on causal reasoning abilities of young children in the GD.

Line 476-479: However, if young children lacked this knowledge, it would have been challenging to pass the task within the 8 trials they were given. To explore this further, 3-year-olds could be provided with additional experience (e.g., poking tissues or paper with sticks) to see if they generalized this knowledge to the experimental context.

We also addressed the development of executive functions as a possible explanation for 3-year-olds' performance in this task. We think the processing demands of our tasks was very low -only keeping track of two events where information can be updated each time the barrier is lifted (e.g., first ripped with sticker, then with pen). We know from over-imitation studies that children of this age are very well able to imitate a sequence of events that shows the ability to track and remember event sequences. However, we did not test this explicitly. We added a sentence to make this clearer.

Line 485-486: In this study, we did not assess children's ability to keep track of two events in the absence of inference-making requirement (but see, 59).

To conclude, again, this is a strong revision and a very nice piece of work.

Thank you very much!

Referee: 2

Comments to the Author(s).

I believe the authors have sufficiently addressed my initial round of comments, as well as the various important issues raised by the other reviewers. I have only one minor follow-up query about the new examples in the introduction, which seem somewhat detached from the phenomenon that the experimental protocol assesses. The authors write: “(e.g., tapping on the nut to infer its fullness or weighing a tool before using it to crack a nut)” – though both of these examples involve the animal making some kind of environmental manipulation in order to gather information. In the experiments here, the animal witnesses a hidden event and is then presented with the opportunity to act upon an inferred causal relationship that took place during that hidden event. Are there naturalistic examples that more closely parallel this structure than the “tapping on the nut” or “weighing a tool” examples? Perhaps I am misunderstanding either the examples or the task itself, and if so, I would appreciate any clarification that the authors could provide.

Thank you very much for pointing this out. It is true that what we are aiming to assess in this paper is whether the subjects can make inferences based on ‘naturalistic’ observations of hidden events rather than learning based on ‘egocentric’ feedback/acting on the objects themselves. That is also why we did not provide our participants with prior experience with poking and ripping objects in advance. In the introduction, we chose these examples with the existing research in mind about capuchin monkeys’ capabilities with regards to hidden causes; however, we agree that these examples do not represent what we are measuring in our task, so we changed them accordingly. We also stressed the difference between learning from naturalistic observations vs. one’s own manipulations in line with AE’s suggestion.

Line 94-99: In such unstable environments, the ability to make inferences about hidden causes based on incomplete observational evidence (e.g., inferring that a nut is empty based on wormholes on its shell) would allow the monkey to go beyond learning only about causal relations that they can manipulate (e.g., tapping on the nut to infer its fullness or weighing a tool before using it to crack a nut) (40).

I think this paper will make a useful and timely addition to the literature on the developmental and evolutionary roots of causal understanding. Thank you for the opportunity to review this interesting research and best wishes moving forward with this line of work.

Thank you very much for the kind words and encouragement!